# Data Provenance Auditing of Fine-Tuned Large Language Models with a Text-Preserving Technique

**Yanming Li** [1 2 3]  **Cédric Eichler** [2 4 1]  **Nicolas Anciaux** [1 2 3]  **Alexandra Bensamoun** [3]  **Lorena Gonzalez-Manzano** [5]
**Seifeddine Ghozzi** [6]

## Abstract

We propose a system for marking sensitive or copyrighted texts to detect their use in fine-tuning large language models under black-box access with statistical guarantees. Our method builds digital "marks" using invisible Unicode characters organized into ("cue", "reply") pairs. During an audit, prompts containing only "cue" fragments are issued to trigger regurgitation of the corresponding "reply", indicating document usage. To control false positives, we compare against held-out counterfactual marks and apply a ranking test, yielding a verifiable bound on the false positive rate. Empirically, we obtain a true positive rate of 96.7% at 0% false positive rate and reply regurgitation rates exceeding 28% per document with only 40 (4%) watermarked documents. The approach is minimally invasive, scalable across many sources, robust to standard processing pipelines, and achieves high detection power even when marked data is a small fraction of the fine-tuning corpus.

## 1. Introduction

Large language models (LLMs) are trained and fine-tuned using vast collections of text, but the fine-tuning stage, based on task-specific often proprietary corpora, raises particularly salient questions about provenance, as the composition of these datasets is typically not disclosed. Fine-tuning data frequently consists of high-value content, including personal or copyrighted works (e.g., resumes for recruitment models, literary texts for creative assistance), whose limited size

and targeted nature can disproportionately shape a model's downstream behavior. This opacity leaves data owners and authors without effective means to determine whether their content has been used to fine-tune a deployed model or chatbot, creating challenges for privacy, intellectual property, and data governance that legal and policy instruments alone struggle to address. Beyond adversarial settings, provenance guarantees can also be meaningful in trusted deployment scenarios, such as fine-tuning-as-a-service, where a cooperative provider enables post-hoc auditing to verify whether marked documents were included in training.

We consider the following research question: given only black-box access to a fine-tuned LLM, can we reliably determine whether a specific corpus of documents belonging to a given rights holder was included in the fine-tuning set? We assume a minimal access setting with only prompt–response interaction, consistent with typical domain-specific LLM deployments, such as customer-service chatbots obtained through fine-tuning.

Existing technical approaches fall short of providing reliable text-preserving provenance evidence. Training data can sometimes reappear verbatim in model outputs (Carlini et al., 2023), but such occurrences are hard to elicit and do not provide statistically controlled evidence[1]. Membership inference attacks (MIAs) have been increasingly adapted to data provenance in LLMs, but their detection power (Duan et al., 2024; Meeus et al., 2025) and evidentiary reliability (Zhang et al., 2025) remain limited and contested in post-hoc settings. Prior copyright-protection techniques for text (Du et al., 2025) are typically invasive, relying on visible or semantically unnatural text modifications that are incompatible with sensitive or high-quality content.

We introduce a minimally invasive method for auditing fine-tuning data usage that enables statistically sound membership verification. Our approach is based on text-preserving watermarking using invisible Unicode sequences embedded into documents prior to fine-tuning. Each watermark is split into a *cue* and a corresponding *reply*, embedded in

---

[1]Petscraft, Inria, Palaiseau, France [2]INSA CVL, Bourges, France [3]Université Paris-Saclay, Palaiseau, France [4]Université d'Orléans, Orléans, France [5]Universidad Carlos III de Madrid, Madrid, Spain [6]ENSTA, Institut Polytechnique de Paris, Palaiseau, France. Correspondence to: Cédric Eichler <cedric.eichler@inria.fr>.

---

[1]See an external analysis by Nicholas Carlini (2025), which explicitly advises against using these results for copyright litigation (link).

disjoint chunks of text. At audit time, targeted black-box challenges present the *cue*; reproduction of the *reply* in the model's output indicates memorization consistent with training on the marked documents. Detection is formalized as a ranking-based hypothesis test against a reserved set of random counterfactual watermarks, yielding explicit and provable bound on the false-positive rate (FPR).

We evaluate our method across multiple open-source LLMs and text domains, including news and poetry. Experiments examine memorization, sensitivity to fine-tuning set size, multi-watermarks interference, true positive rate (TPR) and FPR. Results show multi-watermark support and reliable detection with no false positives observed in our experiments, and strong detection rates even when the marked documents constitute a small fraction of the fine-tuning data.

Our main contributions are:
(1) A novel, minimally invasive, text-preserving watermarking framework for post-hoc auditing of LLM fine-tuning data, based on invisible canaries and black-box challenges.
(2) A statistically grounded decision procedure using ranking against reserved counterfactual watermarks, providing provable control of the FPR and supporting large-scale, multi-user attribution.
(3) An extensive empirical evaluation across models and text domains demonstrating robustness to scale, high TPR, low FPR, and resilience to multi-watermark interference.

We release code, watermark embedding and detection utilities, and evaluation scripts to facilitate reproducibility[2]

The remainder of the paper is organized as follows. Section 2 formulates the dataset provenance auditing problem and its requirements. Section 3 describes our approach, including watermark design, embedding, detection, and FPR-bounded decision. Section 4 presents experimental results, Section 5 reviews related work, Section 6 discusses limitations, and Section 7 concludes.

## 2. Problem statement

**Dataset provenance auditing (DPA) problem.** We consider the problem of auditing whether a collection of high-quality documents has been used to fine-tune a LLM under the assumption that the documents can be proactively marked before potential usage. The goal is to enable reliable post-hoc detection of unauthorized usage.

We decompose such auditing procedures in two components: (1) a *marking scheme* $mark$, which embeds imperceptible marks in documents prior to release, and (2) a *verification*

---

[2]All code and evaluation scripts are available at https://github.com/liam-0/Data-Provenance-Auditing-of-Fine-Tuned-Large-Language-Models-with-a-Text-Preserving-Technique.

*procedure* $verif$, that queries the suspicious model and makes a decision from the resulting responses.

Let $\mathcal{D}$ denote a protected dataset (e.g., a collection of news articles, poems...), marked prior to release. Let $\mathcal{L}$ denote a suspicious model fine-tuned on a dataset that may include $mark(\mathcal{D})$. The verification procedure has access to both $mark(\mathcal{D})$ and $\mathcal{L}$ and outputs a binary decision. Specifically, $verif(mark(\mathcal{D}), \mathcal{L}) \in \{0, 1\}$, where output 1 indicates the claim that $\mathcal{D}$ (or part of it) was used in the fine-tuning of $\mathcal{L}$, and output 0 indicates no such claim. A valid DPA scheme must satisfy requirements at two levels:

**Threat model.** We consider black-box dataset provenance auditing for fine-tuned LLMs. A data owner proactively marks documents before release and later audits a suspicious fine-tuned model through API access only. The verifier has access to the marked collection and model outputs, but not to model weights, gradients, logits, token probabilities, or training data. We assume the model provider may apply standard preprocessing or normalization during fine-tuning, but does not employ adaptive defenses explicitly designed to subvert provenance auditing. Attacks such as watermark forgery or targeted watermark removal are therefore considered out of scope and discussed as limitations.

**Data marking requirements.** A marking scheme $mark$ embeds marks into documents $D \in \mathcal{D}$ before possible fine-tuning and must satisfy the following:

*Text authenticity.* The marking scheme must preserve the visible text exactly, ensuring that the author's original wording and formatting remain unchanged.

*Scalability.* The marking scheme must support the assignment of many distinct watermarks across users and documents, enabling multi-watermark attribution without collisions or interference.

*Robustness.* Marks must remain detectable despite common data processing and normalization pipelines; robustness is required only against non-adversarial transformation and filters. We discuss robustness to adversarial transformations with comparison to the existing works in limitations and in Appendix F.

**Verification procedure requirements.** Given a suspicious model $\mathcal{L}$ and a set of marked sensitive documents $mark(\mathcal{D})$, the verification procedure $verif$ must satisfy:

*Black-box.* The procedure must operate using only input–output access to $\mathcal{L}$, without reliance on internal model parameters, gradients, or token probabilities. This reflects realistic deployment settings for proprietary or hosted LLMs.

*Soundness.* Any positive claim $verif(mark(\mathcal{D}), \mathcal{L}) = 1$ must be correct except with a provably small probability. The FPR must be strictly bounded, so that positive findings

constitute credible evidence of data usage.

*Completeness.* If a sufficiently large number of marked documents (e.g., tens of documents) was used during fine-tuning, the procedure should detect it with high probability, ensuring that usage cannot systematically evade detection.

# 3. Provenance Auditing through Non-Visible Watermarks

Our proposal is sketched in Figure 1 and described below. Watermarks are generated by pairing a *cue* and a *reply*, both composed of syllables of invisible characters (Section 3.1). When a user requests a watermark, they are attributed a set of $K$ watermarks (Section 3.2), commit to one watermark $w$, and embed it into their sensitive collection prior to release (Section 3.3).

At audit time, a suspicious model is queried via black-box access using (green) chunks containing a *cue*, and the resulting outputs are analyzed to detect the presence of the corresponding *reply* (Section 3.4). This process is repeated using the $K$-1 left-out watermarks, which act as counterfactuals and enable statistically grounded decision: watermarks are ranked according to the number of times their *reply* is detected. If the publication watermark ranks above a fixed threshold $k$, we conclude that sensitive data was included in the model's fine-tuning set (Section 3.5), with false positive rate at most $k/K$. Detailed algorithms and analyses are provided in Appendix A.

## 3.1. Watermark generation

Our watermarks act as canaries, i.e., synthetic sequences designed to be memorized during training and selectively reproduced at inference time. They are constructed from an *alphabet* $\mathcal{A}$ composed of non-rendering characters with no impact on formatting, ensuring text-preservation. In preliminary experiments, isolated invisible characters yielded weak detection. We therefore group invisible characters into short clusters, which we refer to as *syllables*. A syllable is an ordered list of $m$ characters of $\mathcal{A}$, denoted $s \in \mathcal{A}^m$. A *watermark w* is an ordered list of n syllables, $w = (s_i)_{i=1..n}$, $s \in \mathcal{A}^m$.

Each watermark consists of a *cue* and a *reply*, denoted $wc$ and $wr$ respectively, with: $wc = (s_i)_{s_i \in w, i \leq j}$ and $wr = (s_i)_{s_i \in w, i > j}$

We call the *tail of a cue* its last $t$ syllables. The *cue* is embedded in a prompt and serves as the stimulus for the appearance of the *reply*.

We note $W$ the domain of watermarks. To minimize false positives, we aim to reduce the likelihood that a *reply* associated with a dataset is observed by a model not trained on it. To this end, we impose the following constraints on $W$:

(1) Each *cue* corresponds to a unique *reply* and vice versa: $\forall(w_1, w_2) \in W^2, \ (wc_1 = wc_2 \vee wr_1 = wr_2) \Leftrightarrow w_1 = w_2$

(2) No *cue* is contained in a *reply* and vice versa: $\forall(w_1, w_2) \in W^2, \ \neg(wc_1 \sqsubseteq (wr_2)) \wedge \neg(wr_1 \sqsubseteq (wc_2))$ where $x \sqsubseteq y$ denote that the ordered list $x$ is a contiguous subsequence of the ordered list $y$.

Assuming $j \geq n - j$, we have (see Appendix A):

$$|W| \geq \max_{x \in [1, |\mathcal{A}|^{m(n-j)}]} \min(x, |\mathcal{A}|^{mj} - x \times (m(2j - n) + 1)|\mathcal{A}|^{m(2j-n)})$$

In particular, with $n = 2 * j$, $|W| \geq \frac{|\mathcal{A}|^{mj}}{2}$.

## 3.2. Watermark Attribution

We assume the presence of a trusted entity responsible for generating and assigning watermarks while enforcing the constraints defined in Sec. 3.1. This entity guarantees the uniqueness of each watermark by maintaining a record of all previously communicated watermarks, denoted $W_{com}$.

Whenever a user requests a new watermark, the trusted entity samples a set $W_K$ uniformly at random from $W \setminus (W_{com})$, updates its record $W_{com} \leftarrow W_{com} \cup W_K$, and returns $W_K$.

$W_K$ serves as a candidate set of watermarks for the user: they select one uniformly at random from $W_K$ for actual use and retain the remaining watermarks as counterfactuals for the decision procedure (see Sec. 3.5).

## 3.3. Watermark Embedding

A document $D$ is first segmented into an ordered list of text units, $(d_i)_{i=1..\Delta}$. In whitespace-delimited languages, these units can be obtained by splitting on spaces; in languages without explicit word boundaries, they can be obtained using standard language-specific segmentation tools.

Because the *cue* is used to construct prompts in which no *reply* should appear, the two are embedded in separate chunks of the document. However, to encourage the model to associate the tail of the *cue* with the corresponding *reply*, the last $t$ syllable of the *cue* to be embedded alongside the *reply*.

To ensure that 1) significant samples of the document necessarily contains both the *cue* and the *reply*, and 2) that *cue* chunks fit within context window, we chunk $D$ into $\lceil \Delta/\delta \rceil$ sub-documents $(D_i)_{i=1..\lceil \Delta/\delta \rceil}$. Sub-documents with odd indexes, or *cue* chunks, are embedded with the first $j - t$ syllables of the *cue*, i.e. the *cue* minus its tail. Sub-documents with even indexes are called *reply* chunks and embedded with $(s_i)_{s_i \in w, i > j - t}$, i.e., the tail of the *cue* plus the *reply*. Syllables are inserted cyclically every *step* word.

The resulting watermarked document consists of an ordered

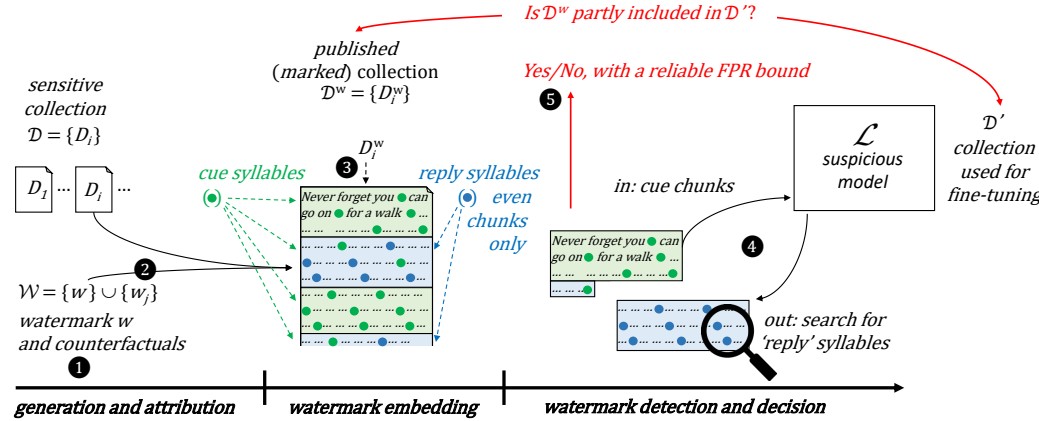

*Figure 1.* Overview of the proposal. A user is attributed a set $W_K$ of $K$ watermarks and commits to one, $w$, which is embedded into their sensitive collection prior to release. At audit time, the model is queried using the *cue* of each watermark of $W_K$, which may elicit the corresponding *reply*. The target watermark $w$ is then ranked against counterfactuals. If it exceeds a fixed threshold $k$, we infer that sensitive data was included in the model's fine-tuning set.

sequence of syllables (each containing $m$ invisible characters) interleaved with the original words, which remain unchanged and in their original order. Appendix A.2 details the embedding algorithm and the number of watermark repetitions it inserts in a document.

### 3.4. Watermark Detection

**Single document.** For a document $D$ watermarked with $w$, denoted $D^w$, verification proceeds by constructing $\lfloor \frac{\lceil \Delta/\delta \rceil}{2} \rfloor$ prompts $ID_i^w$. Each prompt consists of a *cue* chunk concatenated with the beginning of the subsequent *reply* chunk, truncated after its first $t$ invisible syllables so as to complete the *cue*.

The chunk-level verification procedure, described in Algorithm 2, consists in prompting $\mathcal{L}$ $\lambda$ times with $ID_i^w$. It outputs 1 if and only if the sign $wr$ is detected in the output of $\mathcal{L}$ at least once. The document-level verification score of a document $D^w$, is the sum of its chunk-level scores:

$$verif(\mathcal{L}, D^w, \lambda) = \sum_{i=1..\lfloor \frac{\lceil \Delta/\delta \rceil}{2} \rfloor} verif(\mathcal{L}, ID_i^w, \lambda)$$

**Collection of documents.** The score of an entire collection of documents $\mathcal{D}$ embedded with the same watermark $w$, denoted $\mathcal{D}^w$, is the sum of each document's score:

$$verif(\mathcal{L}, \mathcal{D}^w, \lambda) = \sum_{D^w \in \mathcal{D}^w} verif(\mathcal{L}, D^w, \lambda)$$

### 3.5. Decision process

Given a watermarked collection $\mathcal{D}^w$, we compare its score given by *verif($\mathcal{L}, \mathcal{D}^w, \lambda$)* to the score of counterfactuals, i.e. watermarks that were never used and cannot have been

used in fine-tuning. Since $w$ is randomly sampled from a candidate set $W_K$, left-out elements may serve as counterfactuals. Importantly, a watermark cannot be scored *ex nihilo*: the scores of the counterfactual watermarks must be computed in the same context as the one being tested. Thus, *verif($\mathcal{L}, \mathcal{D}^w, \lambda$)* is ranked compared to *verif($\mathcal{L}, \mathcal{D}^{w_K}, \lambda$)* for each $w_K \in W_K \backslash \{w\}$. Membership is claimed for a watermark if its score is $> 0$ and its rank $\geq k$ for a given $k$. The procedure is given in Algorithm 3.

While not providing an auditing approach, Zhang et al. (2025) discussed the properties they should verify to obtain sound and tractable membership proofs. In particular, they demonstrated that the FPR of the ranking test is bounded if: (1) target data ($w$) is explicitly sampled uniformly at random from a candidate set ($W_K$) whose left-out members serve as counterfactuals; (2) the injected data does not impact other data published afterwards, e.g. if it carries no useful information. Since our proposed watermarks are invisible, they are unlikely to impact the production of subsequent documents by third parties. Hence, our approach satisfies these conditions and, in line with (Zhang et al., 2025), we have FPR $\leq \frac{k}{K}$.

### 3.6. Practical Considerations

Certain requirements of Section 2 are achieved by design: **text authenticity** follows from the use of non-rendering Unicode which leaves the visible text sequence intact; **black-box operability** stems from the verification requiring text I/Os only; **scalability** stems from the exponential growth of the watermark space; and **soundness** is ensured by the ranking test. In practice, we record a cryptographic commitment to $w$ at assignment time, which prevents post-hoc cherry-picking. Beyond these design properties, several practical

aspects must be assessed empirically:

**Choice for the threshold** $k$**.** The ranking test gives a simple theoretical rule (FPR $\leq k/K$). The rate at which left-out watermarks are regurgitated must be verified in practice.

**Completeness and verification budget (in I/Os).** Completeness depends on the per-chunk *hit* probability $p$ of a single chunk-level verification success. Repeating the prompt $\lambda$ times reduces the per-chunk *miss* probability to $(1-p)^\lambda$. Probing $X$ independent chunks from the same collection, the collection-level probability of having no *hit* is $p_{FN} = (1-p)^{\lambda X}$. Estimating realistic values of $p$ across models and domains and instantiating $(\lambda, X)$ to achieve practical error budgets, requires empirical measurement.

**Scalability in practice.** While the watermark space $|W|$ grows exponentially in theory, its effective use must be quantified, including collision risk and attribution accuracy under multi-watermark attribution.

**Robustness.** Appendix E empirically demonstrates that watermarks are robust to common non-adversarial transformations they: (1) survive to insertion and recovery from 4 websites known to be scrapped for data collection such as Wikipedia and GitHub; (2) withstand 6 standard data cleaning and processing pipelines, including C4 (Raffel et al., 2020), CCNet (Wenzek et al., 2020), the Pile (Gao et al., 2020), RedPajama (Weber et al., 2024), Dolma (Soldaini et al., 2024) and FineWeb (Penedo et al., 2024); (3) are reliably recognized by 10 representative tokenizers (including tokenizers used by Mistral-7B-v0.1, DeepSeek-R1 and gpt-4o); and (4) are not filtered by safety mechanisms in public chatbot interfaces like ChatGPT, Le Chat, and DeepSeek.

## 4. Experimental Evaluation

This section presents the experimental evaluation of our proposed approach. We focus on its detection power, measured as the average regurgitation rate of a *reply* under varying conditions. Specifically, we investigate how regurgitation depends on (i) the size of the watermarked collection, (ii) the total fine-tuning dataset size, and (iii) the number of unique watermarks. We further show that spurious regurgitation is negligible, yielding high TPR@0%FPR even in challenging settings where the closest existing baseline fails.

### 4.1. Experimental settings

**Datasets.** We evaluate our method on three datasets representative of personal and copyrighted textual content. (1) **Blog1k**, a subset of the Blog Authorship Corpus, containing*personal blog* posts. (2) **Poems**, containing *poems* collected from the Poetry Foundation. (3) **News** (CNN/DailyMail) containing longer news articles, enabling evaluation in long-context generation settings. Entries

shorter than 200 words are discarded, reducing Blog1k to 17,843 documents with an average length of 360.03 words, Poems to 5,636 documents of 480.01 words on average, and News to 11,837 documents averaging 1,519.96 words.

**LLMs** $\mathcal{L}$**.** All experiments are conducted using LLaMA-2-7b-hf and Mistral-7B-v0.1 (denoted as LLaMA and Mistral, respectively), two prominent open-source models, ensuring reproducibility. We limit each output to 200 tokens and detail the used parameters in Appendix B.

**Parametrization of the approach.** The number $m$ of characters per syllable entails a trade-off. If $m$ is too small, a syllable may be disregarded as noise; if it is too large, its final part may cease to be associated to preceding "legitimate" words, thereby diminishing regurgitation rates. Based on empirical observations, we set $m = 4$.

Similarly, $n$, the size of the watermark, must be appropriately calibrated to balance the trade-off between legitimate and spurious detection. If $n$ is too small, the watermark is more likely to occur "by chance", whereas if it is too large, even minor deviations may prevent successful detection. Based on empirical evidence, we set $n=8$. We choose $j=5$ and $t=1$ so that four syllables are embedded repeatedly in each document chunk, irrespective of parity.

The alphabet comprises 121 invisible Unicode characters, yielding $|W| \geq 10^{24}$. The selection process is described in Appendix E.1.

For the Blog1k and Poems datasets, given a document $D$ of length $\Delta$, we set $\delta = \lceil \Delta/2 \rceil$ and partition $D$ into two consecutive chunks. For News, we set $\delta = 200$, each $D$ being partitioned in approximately 8 chunks.

**Evaluation and default values.** We report the probability of successful chunk-level verification at $\lambda = 1$ with 95% confidence intervals. For each chunk, regurgitation is averaged over five watermarks and three prompts each, and the final probability is obtained by averaging over all chunks.

Unless otherwise stated, 1) datasets are downsampled to 1,000 entries for fine-tuning, with randomly selected subsets watermarked to form the sensitive collections; 2) the number of watermarked documents is 40 across all models and datasets, except for Mistral on Blog1k (60), following subsection 4.2 to target intermediate regurgitation regimes. Appendix C reports detailed dataset statistics.

### 4.2. Regurgitation scales with the size of the watermarked collection

To evaluate how regurgitation scales with the size of the watermarked collection, we embed a single watermark $w$ into a subset $\mathcal{D}^w$ of the fine-tuning dataset. For Blog1k and Poems, $|\mathcal{D}^w|$ ranges from 10 to 100 documents (1-10%), and for News, which contains longer documents split into

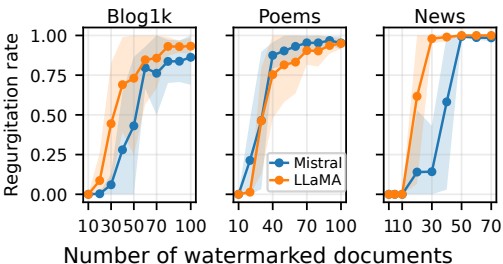

*Figure 2.* Average regurgitation rate as a function of watermarked collection size. Shaded areas indicate 95% confidence intervals.

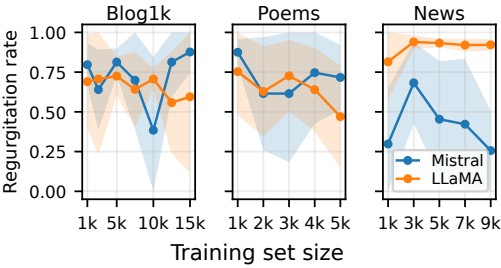

*Figure 3.* Average regurgitation rate as a function of fine-tuning dataset size. Shaded areas denote 95% confidence intervals.

more chunks, from 1 to 70 documents (0.1-7%). Figure 2 reports the chunk-level regurgitation rates.

Regurgitation increases monotonically with $|\mathcal{D}^w|$. For small watermarked subsets, the rate remains near zero; beyond a model- and dataset-dependent threshold, it increases sharply before saturating between 0.75 and 0.98. For instance, on Blog1k, this transition occurs at $|\mathcal{D}^w| \approx 40$ for LLaMA and $|\mathcal{D}^w| \approx 60$ for Mistral. Despite these differences, both models converge to a similar saturation level of approximately 0.75 by $|\mathcal{D}^w| = 70$. Table 1 summarizes the corresponding hit and miss probabilities at the collection level.

**Completeness.** This experiment provides an empirical estimate of the hit probability $p$ for a single prompt and the miss probability $p_{FN}$ over a collection. For both models and all datasets, $p_{FN} < 10^{-2}$ with 30 watermarked documents, decreasing to as low as $10^{-59}$, confirming that watermarks remain highly detectable even at moderate collection sizes.

### 4.3. Regurgitation is not systematically reduced by increasing the number of unmarked documents

To study the effect of fine-tuning dataset size, we fix $|\mathcal{D}^w|$ and vary the amount of unmarked data. Across settings, the proportion of watermarked documents ranges from a maximum of 6% down to a minimum of 0.25% of the dataset. Figure 3 reports chunk-level regurgitation rates.

Across all datasets and models, increasing the amount of unmarked document does not produce a monotonic reduction in regurgitation. Instead, rates fluctuates within a relatively stable range. On Blog1k and Poems, both models exhibit non-monotonic behavior as the fine-tuning set grows, with Mistral showing higher variability. On News, LLaMA maintains consistently high regurgitation across all training scales, whereas Mistral shows pronounced fluctuations, peaking at 0.74 with 3,000 fine-tuning documents, compared to approximately 0.25 at 1,000 and 9,000.

We hypothesize that this variability primarily stems from the relatively small fine-tuning datasets, which may amplify sensitivity to stochastic effects such as data sampling and optimization noise. Consequently, the effect of fine-tuning set size is difficult to disentangle from this variability.

**Scalability w.r.t. finetuning dataset**. Increasing the number of unmarked documents does not yield predictable changes in regurgitation, which remains largely non-monotonic. The fraction of watermarked documents alone is a poor predictor of regurgitation strength.

### 4.4. Verification support multi-watermark settings

We investigate the impact of the number of distinct watermarks present in the fine-tuning dataset, denoted $U$, on the verification procedure. Unlike previous experiments, where the five watermarks were embedded in the same collection and evaluated independently, each watermark is embedded into a disjoint subset of the dataset. For $U = 1$, watermarks are evaluated in isolation, and are all present in the fine-tuning dataset starting at $U = 5$, set to 5,000 documents to accommodate up to 50 unique watermarks. Figure 4 shows the aggregated chunk-level regurgitation rate. Detailed per-watermark results are provided in Appendix D.

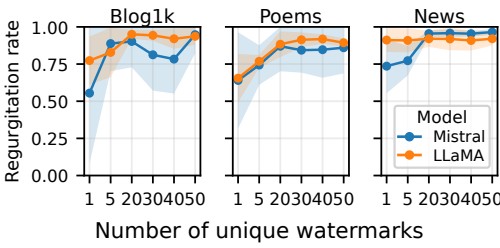

*Figure 4.* Average regurgitation rate as a function of the number of unique watermarks in the fine-tuning dataset. Shaded areas denote 95% confidence intervals.

As $U$ increases from 1 to moderate values ($U = 5$ and $U = 20$), regurgitation rises across all datasets before stabilizing near saturation for both models (approximately 0.85–0.95 on Blog1k and Poems, and above 0.9 on News). This trend is particularly pronounced for Mistral, whose regurgitation increases from roughly 0.5–0.7 at $U = 1$ to above 0.85

*Table 1.* Empirical estimation of chunk hit probability $p$ and collection miss probability $p_{FN}$.

*(a)* Blog1k

| nbdocs | Mistral | | LLaMA | |
|---|---|---|---|---|
| | $p$ | $p_{FN}$ | $p$ | $p_{FN}$ |
| 30 | 0.060 | 0.156 | 0.444 | $2.2 \times 10^{-8}$ |
| 40 | 0.280 | $2 \times 10^{-6}$ | 0.690 | $4.5 \times 10^{-21}$ |
| 50 | 0.431 | $5.9 \times 10^{-13}$ | 0.731 | $3.3 \times 10^{-29}$ |

*(b)* Poems

| nbdocs | Mistral | | LLaMA | |
|---|---|---|---|---|
| | $p$ | $p_{FN}$ | $p$ | $p_{FN}$ |
| 30 | 0.464 | $7.3 \times 10^{-9}$ | 0.462 | $8.3 \times 10^{-9}$ |
| 40 | 0.875 | $7.5 \times 10^{-37}$ | 0.753 | $4.8 \times 10^{-25}$ |
| 50 | 0.903 | $2.6 \times 10^{-51}$ | 0.815 | $2.5 \times 10^{-37}$ |

*(c)* News

| nbdocs | Mistral | | LLaMA | |
|---|---|---|---|---|
| | $p$ | $p_{FN}$ | $p$ | $p_{FN}$ |
| 30 | 0.039 | $8.6 \times 10^{-3}$ | 0.677 | $\approx 10^{-59}$ |
| 40 | 0.298 | $\approx 10^{-25}$ | 0.815 | $\approx 10^{-117}$ |
| 50 | 0.772 | $\approx 10^{-129}$ | 0.898 | $\approx 10^{-199}$ |

at $U = 50$ across all datasets, indicating that exposure to multiple distinct watermarks does not diminish, and may even strengthen, memorization.

To examine whether this behavior generalizes to more recent and larger-scale models, we extend the analysis to GPT-OSS-20B (Appendix D) and observe the same trend. Fine-tuning parameters and watermarked-text statistics are reported in Appendices B and C.

> **Scalability: multi-watermark support.** Across all datasets and models, increasing the number of distinct watermarks does not reduce regurgitation rates; detection performance remains stable or improves under multi-watermark fine-tuning.

### 4.5. Decision yields high TPR at low FPR.

While previous experiments focused on regurgitation rates, we now evaluate the *decision procedure* for assessing membership following Algorithm 3. Let $w_{i \in [1,5]}$ be the 5 watermarks used throughout the experiments, and let $\bar{w}_{j,j \in [1,99]}$ be randomly generated counterfactual watermarks. For each dataset, each $w_i$ is scored using $verif(\mathcal{L}_i, \mathcal{D}^{w_i}, 1)$, where $\mathcal{L}_i$ denotes the model $\mathcal{L}$ fine-tuned on the dataset containing $\mathcal{D}^{w_i}$. Counterfactual watermarks are evaluated via $verif(\mathcal{L}_i, \mathcal{D}^{\bar{w}_j}, 1)$ with $i \in [1,5]$ selected at random. Table 2 reports the resulting scores over three repetitions.

*Table 2.* Average scores over three ranking tests. Legitimate watermarks ($w_i$) yield substantial regurgitation, while counterfactuals ($\bar{w}_j$) produce none.

| Model Fine tuned on $\mathcal{D}^{w_i}$ | Dataset ($|\mathcal{D}^{w_i}|$) | Scores on $w_{i,i \in [1,5]}$ | | | All $\bar{w}_j$ |
|---|---|---|---|---|---|
| | | Avg. | Worst $w_i$ | Best $w_i$ | $j \in [1,99]$ |
| Mistral-7B | Blog1k (60) | 48 | 39 | 57 | 0 |
| | Poems (40) | 35.2 | 32 | 38 | 0 |
| | News (40) | 51.9 | 0 | 110.72 | 0 |
| LLaMA-2-7B | Blog1k (40) | 27.6 | 16 | 35.2 | 0 |
| | Poems (40) | 30 | 0.37 | 36 | 0 |
| | News (40) | 141.86 | 86.5 | 169.54 | 0 |

No spurious regurgitation (i.e. regurgitation of the *reply* of a counterfactual watermark) is detected over more than $10^5$ challenges. In contrast, legitimate watermarks generally exhibit substantial regurgitation, with average rates above 0.5. A single exception occurs for Mistral fine-tuned on News, where one watermark yields 0 regurgitation. Overall, these results correspond to an approximate TPR of 96.7% at 0% FPR with k=1.

**96.7%@0%FPR**. Spurious detections are negligible, suggesting that a small k=1 can be safely adopted. Additionally, verification may be performed on a subset of the sensitive collection to reduce I/O costs without compromising detection.

### 4.6. Closest existing work fails under multi-watermark evaluation

To the best of our knowledge, the homoglyph watermarking scheme of (Wei et al., 2024) is the only existing text-preserving watermarking approach (see Section 5). We evaluate this baseline under the setting of Section 4.4 comprising 50 unique watermarks.

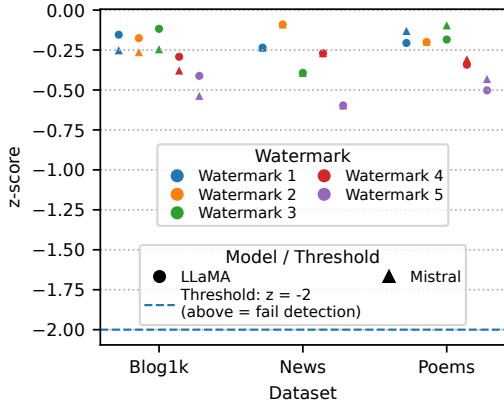

*Figure 5.* Z-scores for the homoglyph watermarking baseline (Wei et al., 2024). Each point corresponds to one watermark. The dashed line indicates the significance threshold ($p \approx 0.05$) above which score fails to achieve statistically significant detection.

The baseline relies on average token loss and empirical Z-scores computed from a null distribution of 100 random sequences. Figure 5 shows the resulting Z-scores, which are concentrated around $Z \approx -0.1$ to $-0.6$ corresponding to statistically non-significant detection ($p > 0.2$). This indicates that the baseline fails to reliably distinguish watermarked samples from the null distribution in this regime.

In contrast, our method produces reliable membership signals across all datasets and models, exhibiting regurgitation rates $> 0.8$ under the same setting (see Section. 4.4).

*Table 3.* DPA techniques for free text generation and their capacity to preserve the text (Auth.), operate under a black-box model (BB), and provide statistically grounded proofs (Sound).

| Approaches | Auth. | BB | Sound |
|---|---|---|---|
| Non-intrusive MIAs, e.g. (Duarte et al., 2024; Puerto et al., 2025) | ✓ | Some | ✗ |
| *Waterfall* (Lau et al., 2024) | ✗ | ✓ | ✗ |
| *STAMP* (Rastogi et al., 2025) | ✗ | ✗ | ✓ |
| Fictuous knowledge (Cui et al., 2025) | ✗ | ✓ | ✓ |
| Copyright traps (Meeus et al., 2024) | ✗ | ✗ | ✗ |
| Hash (Wei et al., 2024) | ✗ | ✗ | ✓ |
| Homoglyphs (Wei et al., 2024) | ✓ | ✗ | ✓ |
| Ours | ✓ | ✓ | ✓ |

**Baseline detection fails in challenging regimes.** Across all datasets and models, the homoglyph watermarking scheme (Wei et al., 2024) fails to produce statistically significant detection under multi-watermark evaluation. In contrast, our method consistently yields strong and reliable membership signals.

# 5. Related work

## 5.1. Generative AI, watermarking, and DPA

Prior work on watermarking in generative AI largely assumes that the model provider controls the watermarking process (Zhao et al., 2025a). Most approaches focus on watermarking model outputs to detect AI-generated content, to protect model copyright, where watermarks are used to trace content or derived models back to a specific source-model (Zhao et al., 2023; He et al., 2022; Zhao et al., 2022), or to associate generated text with sources in their training data (Lu et al., 2025). We focus on dataset watermarking for DPA, whose goal is to enable a data owner to detect and *prove* unauthorized use of their data during model training or adaptation. As noted by a recent survey (Du et al., 2025), "insufficient attention has been paid to the dataset copyright issues posed by LLMs". Existing DPA techniques predominantly focus on the image domain (e.g., Wenger et al., 2024; Chen & Pattabiraman, 2025), or on classification tasks, where provenance is inferred from predicted labels (e.g., Choquette-Choo et al., 2021; Li et al., 2022; Liu et al., 2024; Tang et al., 2023).

## 5.2. DPA in LLMs for free text generation

DPA methods are commonly categorized by their degree of intrusiveness (Chen & Pattabiraman, 2025; Du et al., 2025).

**Non-intrusive DPA** methods aim to detect usage without modifying the original data, typically using decision-boundary analysis for classification task or model-based analysis (Du et al., 2025). The latter largely overlaps with MIAs (Shokri et al., 2017), a form of privacy attack increasingly adapted to DPA (Zhang et al., 2025). However, the effectiveness of such approaches on free-text generation has been increasingly questioned (Duan et al., 2024; Meeus

et al., 2025). While some recent work show promising detection (Duarte et al., 2024) in particular at the dataset level (Maini et al., 2024; Puerto et al., 2025), the difficulty of sampling from the null hypothesis questions their ability to provide sound training-data proofs (Zhang et al., 2025).

**Intrusive DPA** techniques embeds detectable signals into the dataset. Recent work has leveraged the radioactivity of model output watermarking techniques (Sander et al., 2024; Zhao et al., 2025b) to adapt them for DPA (Lau et al., 2024; Rastogi et al., 2025). Such approaches may introduce distortions or errors (Lau et al., 2024; Rastogi et al., 2025).

Other approaches rely on the insertion of canaries, such as fictitious knowledge and incorrect facts (Cui et al., 2025) or LLM-generated "copyright traps" (Meeus et al., 2024). Such canaries are perceptible, and it may be unacceptable to insert incorrect knowledge or repeated long sequences in a text. Moreover, copyright traps (Meeus et al., 2024) do not offer statistically grounded guarantees (Zhang et al., 2025; Wei et al., 2024).

Wei et al. (2024) perturbs documents with hash sequences or Unicode homoglyph substitutions, enabling hypothesis testing to verify whether a collection was used in LLM pretraining. To the best of our knowledge, the homoglyph-based approach is the only text-preserving (although some homoglyphs remain visually distinguishable) DPA technique supporting statistically grounded proofs. As with our method, it is not robust to adversarial transformations; however, unlike our approach, it requires word probabilities and does not support multiple concurrent watermarks.

# 6. Discussion and Limitations

While our framework provides sound guarantees of authenticity, black-box deployability, and provable false-positive control, several limitations and broader considerations remain, which could be considered future work.

**Robustness to active removal.** Our threat model assumes non-adversarial preprocessing, excluding active watermark removal. As shown in Appendix E, the proposed scheme is largely preserved under realistic data-collection, preprocessing, tokenization, and public-interface workflows, supporting use cases in which no adversarial removal step is applied. However, the method is inherently brittle to an adversary aware of the watermark, who can apply simple targeted removal strategies such as stripping zero-width characters or filtering suspicious Unicode patterns. This limitation is shared by prior text-watermarking schemes for provenance and copyright protection (Du et al., 2025; Wei et al., 2024).

We further evaluate robustness under untargeted, generic removal attacks in Appendix F, and find that the method is resilient to a broad class of low-cost, semantics-preserving

perturbations, and substantially more robust than prior work. In contrast, it remains vulnerable to high-cost attacks that leverage external models to substantially rewrite the text prior to training.

Importantly, such attacks affect completeness rather than soundness: they may reduce detection power and increase false negatives, but they do not compromise the false-positive control of the ranking-based test. We leave systematic adversarial hardening to future work.

**Robustness to forgery and replay attacks.** The protocol mitigates ex-post watermark forgery through the candidate-set and commitment mechanism. The used watermark is sampled from a candidate set before auditing, unused candidates serve as counterfactuals in the ranking test, and a commitment to the selected watermark is recorded at assignment time. This prevents constructing or cherry-picking a watermark after observing the suspicious model. A stronger contamination or replay attack remains possible if an adversary deliberately reuses a known watermark, or injects it into unrelated data. We emphasize that, as in the rest of this work, we do not aim to provide guarantees under an active adversary. This setting is outside the scope of the current protocol and remains an important direction for future work.

**Trusted third party (TTP) considerations.** We rely on a TTP to generate (*cue*,*reply*) pairs obeying the global uniqueness and overlap constraints, to allocate candidate sets uniformly, and to keep public commitments to protect against cherry-picking. This simplifies soundness and prevents accidental collisions across users, but it introduces trust and operational burdens. Certain directions could reduce that burden like resorting to append-only public logs.

**Scope and training-strategy dependency.** Our method is designed for provenance in supervised fine-tuning, where the $cue \rightarrow reply$ signal concentrates in adapted parameters and supports practical black-box detection. We therefore study adapter-based fine-tuning (Hu et al., 2022) as a representative deployment setting. Other paradigms – such as large-scale pretraining, reinforcement learning from human feedback (Christiano et al., 2017; Ouyang et al., 2022), or some full fine-tuning procedures– may attenuate idiosyncratic signals and reduce detection power, and are not claimed to be covered. Exploring these regimes constitutes natural next steps.

**Legal and ethical considerations.** Proposed marks are imperceptible, but their insertion alters the raw byte representation of a text, which may be unacceptable in certain areas (e.g., for official electronic documents). Furthermore, the probative value of watermark-based audits in litigation will depend on the evolution of legal standards of evidence. On this point, the technical results may well encourage judges to reason by presumption in order to facilitate the

proof of unauthorised use of content, particularly content protected by copyright. In any case, it would be appropriate for the technical audit to be supplemented by governance frameworks and explicit consent mechanisms.

## 7. Conclusion

We presented a minimally invasive, text-preserving framework for dataset provenance auditing of fine-tuned LLMs, embedding invisible Unicode canaries in a *cue –reply* structure and detecting them via black-box prompting. The decision is statistically grounded, with a formally bounded FPR.

Across multiple LLMs and datasets, the approach reliably supports multi-watermark attribution and achieves high TPR@0%FPR with only tens of watermarked documents constituting as little as 0.25% of the total fine-tuning dataset. These results highlight the approach's practicality for provenance tracking in realistic fine-tuning scenarios.

## Author Contributions

This work was carried out as a collaborative effort combining technical, methodological, legal, and security perspectives. Yanming Li contributed to the software development and experiments. Cédric Eichler contributed to the methodology, experimental validation, and writing. Nicolas Anciaux contributed to the conceptualization and scientific coordination of the project, as well as to the writing. Alexandra Bensamoun contributed to the legal analysis and to the framing of the topic. Lorena Gonzalez Manzano contributed to the security analysis. Seifeddine Ghozzi contributed to the preliminary experiments. All authors contributed to discussions and reviewed the manuscript.

## Acknowledgements

This research was supported by grants ANR-22-PECY-0002 (IPoP project), by the DATAIA Convergence Institute as part of the Investments for the Programme d'Investissement d'Avenir (ANR-17-CONV-0003) managed by Inria Saclay, and by PID2023-150310OB-I00 (MORE4AIO project) funded by MCIU, AEI and FEDER. We also thank Inria and U3CM for the PETSAI associated team grant which supports collaborative work between Petscraft and COSEC.

## Impact Statement

Potential societal consequences of our work might be on the juridical side. Indeed, the need for technological support is more evident than the law's inability to ensure legal certainty in a copyright protection context. While there is indeed an exception at European level for text and data min-

ing that suspends the application of copyright and related rights, which could perhaps benefit model training, this is subject to the dual condition that the AI provider has lawful access to the content and that the rights holder has not exercised his opt-out right. However, the opacity surrounding these activities makes it impossible to know whether the conditions are being met. Therefore, the AI Act[3] imposed a transparency obligation on AI providers, which consists of developing and making available to the public a "sufficiently detailed summary" of the content used, for which a model was provided by the AI Office[4]. However, it is not certain that the required level will enable rights holders to enforce their rights. The situation is no more satisfactory in the United States, where some 50 lawsuits are pending on this issue and where the fair use exception is very uncertain. As proof, Anthropic agreed to settle a lawsuit by offering to pay $1.5 billion in compensation for the unauthorized use of 500,000 books[5]. These uncertainties illustrate that legal instruments alone are unlikely to provide sufficient protection or enforceability, reinforcing the need for complementary technical solutions, such as the one presented in this paper.

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

# A. Detailed algorithms and analysis

This section details algorithms applied in this proposal. The notations used in the paper are recalled in table 4.

*Table 4.* Notations

| | |
|---|---|
| $\mathcal{D} = \{D\}$ | A sensitive collection of documents |
| $D = (d_i)_{i=1..\Delta}$ | A document of $\Delta$ words |
| $\mathcal{L}$ | A suspicious model |
| $\mathcal{P} = (verif, mark)$ | A dataset provenance auditing procedure |
| $\mathcal{A}$ | The alphabet of invisible characters |
| $s \in \mathcal{A}^m$ | A syllable of $m$ invisible character |
| $W$ | The domain of watermarks |
| $W_K \subset W$ | Set of $K$ candidate watermarks provided upon request |
| $w = (s_i)_{i=1..n}$ | A watermark of $n$ syllables |
| $wc = (s_i)_{s_i \in w, i \leq j}$ | The "cue" of $w$ made of its first $j$ syllables |
| $(s_i)_{s_i \in w, j-t < i \leq j}$ | The tail of the "cue", composed of its $t$ last syllables |
| $wr = (s_i)_{s_i \in w, i > j}$ | the "reply" of $w$ made of its last $n - j$ syllables |
| $step$ | The "step" in the number of words used in the marking scheme |
| $D = (D_i)_{i=1..\lceil \Delta/\delta \rceil}$ | $D$ is a sequence of contiguous chunks of $\delta$ words |
| $\{D_{2i+1}^{wc}\}$ | Odd chunks of $D^w$ watermarked with syllables of cue not in its tail |
| $\{D_{2i}^{wr}\}$ | Even chunks of $D^w$ watermarked with syllables of reply and of the tail of cue |
| $D^w = mark(D, w)$ | $D$ chunks marked by scheme $mark$ using $w$ |
| $ID_i^w$ | Challenge constructed from $D_{2i-1}^{cw}$ |
| $\lambda$ | Number of repetitions of each $ID_i^w$ during $verif$ |

## A.1. Number of admissible watermarks

We assume $j \geq n - j$, i.e., that the *cue* is longer than the *reply*. A given sequence of $mj$ characters appears in $(m(2j - n) + 1) \times |\mathcal{A}|^{m(2j-n)}$ sequence of $m(n - j)$ characters at most.

For instance, with $j = 2$, $n - j = 1$, and $m = 2$, a sequence of 2 characters, e.g. "aa", may appear in $(m(2j - n) + 1) = 3$ different positions in a sequence of 4 characters, e.g. "aa**", "*aa*", or "**aa". There remains $m(2j - n) = 2$ other characters that may take any value in $|\mathcal{A}|$.

Hence, considering a particular *reply* forbids the usage of at most $(m(2j - n) + 1) \times |\mathcal{A}|^{m(2j-n)}$ distinct *cue*. This is an upper bound as multiple occurrences of a *reply* in a single *cue* are counted multiple time, e.g. "aaaa" is counted twice.

Hence, due to constraint (2), each used *reply* "forbids" at most $(m(2j - n) + 1) \times |\mathcal{A}|^{m(2j-n)}$ *cue*.

Because of condition (1), the number of distinct watermarks equals the minimum of the number of distinct *cue* and *reply*. Hence, we have,

$$|W| \geq max_{x \in [1, |\mathcal{A}|^{m(n-j)}]} min(x, |\mathcal{A}|^{mj} - x \times (m(2j - n) + 1)|\mathcal{A}|^{m(2j-n)})$$

Note that, if $j = n - j$, i.e., the *cue* and *reply* have the same length, (2) reduces to an inequality and the optimum is reached when the space is equally partitioned between *reply* and *cue*. As expected, by substitution in the above equation, we get:

$$|W| \geq max_{x \in [1, |\mathcal{A}|^{mj}]} min(x, |\mathcal{A}|^{mj} - x\}) \quad \text{i.e. } |W| \geq \frac{|\mathcal{A}|^{mj}}{2}$$

## A.2. Embedding

Algorithm 1 presents the embedding process. It chunks $D$ into $\lceil \Delta/\delta \rceil$ sub-documents $(D_i)_{i=1..\lceil \Delta/\delta \rceil}$ containing $\delta$ words (except for the last). Alternatively, a chunk is embedded with either (most of) the *cue*, $(s_i)_{s_i \in w, i \leq j}$, or $(s_i)_{s_i \in w, i \geq j-t}$ (i.e. the $t$ last syllables of the *cue* and the *reply*). Sub-documents with odd indexes, or *cue* chunks, are denoted $\{D_{2i+1}^{wc}\}$. Sub-documents with even indexes or *reply* chunks, are denoted $\{D_{2i}^{wr}\}$.

The function *chunk_mark* embeds the relevant syllables within a subset of the document. The first syllable is inserted after the first word, and the remaining ones are inserted every *step* word. The $j - t$ or $n - j + t$ syllables are injected cyclically throughout the list. When the end of this list is reached, a syllable is not necessarily inserted even if the last one was injected exactly *step* words before. Rather, if the final repetition is incomplete, the remaining syllables are concatenated at the end to ensure full representation.

---

**Algorithm 1** *mark*, embed Watermark $w$ in $D$

---

**Input:** $D = (d_i)_{i=1..\lceil \Delta/\delta \rceil}, \delta, t, step, j, w$
**Output:** $D^w$, $D$ **watermarked with** $w$
 1: $D^w \leftarrow$ empty list
 2: $x \leftarrow 1$ {Start of chunk}
 3: **while** $x + \delta < \Delta$ **do**
 4:     $y \leftarrow x + \delta - 1$
 5:     $D^{wc} \leftarrow$ chunk_mark$((d_i)_{i=x..y}, (s_i)_{s_i \in w, i \leq j-t})$
 6:     Concatenate $D^w$ and $D^{wc}$
 7:     $x \leftarrow x + \lambda$
 8:     $y \leftarrow min(x + \delta - 1, \Delta)$
 9:     $D^{wr} \leftarrow$ chunk_mark$((d_i)_{i=x..y}, (s_i)_{s_i \in w, i > j-t})$
10:     Concatenate $D^w$ and $D^{wr}$
11:     $x \leftarrow x + \delta$
12: **end while**
13: **if** $x \neq \Delta$ **then**
14:     Concatenate $D^w$ and $(d_i)_{i=x..\Delta}$
15: **end if**
16: **return** $D^w$
**Subroutine** *chunk_mark*
**Input: Words** $(d_i)_{i=1..N}$**, syllables** $(s_i)_{i=1..M}$
**Output: Sequence $D^w$ of words and syllables**
17: $D^w \leftarrow$ empty list
18: Append $d_1$ to $D^w$
19: Append $s_1$ to $D^w$
20: $x \leftarrow 2$ {word count}
21: $y \leftarrow 2$ {syllable count}
22: **while** $x + step \leq N$ **do**
23:     Concatenate $D^w$ and $(d_i)_{i=x..x+step-1}$
24:     Append $s_y$ to $D^w$
25:     $x \leftarrow x + step$
26:     $y \leftarrow (y \bmod M) + 1$ {Cycle through syllables}
27: **end while**
28: Concatenate $D^w$ and $(d_i)_{i=x..N}$
29: **while** y $\neq 1$ **do**
30:     Append $s_y$ to $D^w$
31:     $y \leftarrow (y \bmod M) + 1$
32: **end while**
33: **return** $D^w$

---

Following the strategy described in Algorithm 1, if $y$ is the number of syllables taken as input and $x$ is the number of words, $1 + \lfloor \frac{x-2}{step} \rfloor$ are embedded first. Then, any remaining syllables are appended at the end to ensure complete representation.

The total number of repetitions in a sub-document is thus:

$$\text{sub-repetitions(x,y)} = \left\lceil \frac{1 + \left\lfloor \frac{x-2}{step} \right\rfloor}{y} \right\rceil$$

Recall that odd subdocuments are embedded with the $j - t$ first syllables of the *cue*, the remaining $t$ being embedded in even subdocuments alongside the $n - j$ syllables of the *reply*. A document of size $\Delta$ has:

- $\lceil \Delta/\delta \rceil/2$ sub-document of size $\delta$ embedded with a signal of $j - t$ syllables,

- $\lceil \Delta/\delta \rceil/2$-1 sub-documents of size $\delta$ embedded with a signal of $n - j + t$ syllables, and

- a final sub-document of size $\Delta - \delta \cdot (\lceil \Delta/\delta \rceil - 1)$ embedded with a signal of $n - j + t$ syllables.

Thus, the number of repetitions of $(s_i)_{s_i \in w, i \leq j-t}$ in $D$ is:

$$\lceil \Delta/\delta \rceil/2 \times \left\lceil \frac{1 + \left\lfloor \frac{\delta-2}{step} \right\rfloor}{j - t} \right\rceil$$

And the number of repetitions of $(s_i)_{s_i \in w, i > j-t}$ in $D$ is:

$$\lceil \Delta/\delta \rceil/2 \times \left\lceil \frac{1 + \left\lfloor \frac{\delta-2}{step} \right\rfloor}{n - j + t} \right\rceil + \left\lceil \frac{1 + \left\lfloor \frac{\Delta-\delta \cdot (\lceil \Delta/\delta \rceil-1)-2}{step} \right\rfloor}{n - j + t} \right\rceil$$

### A.3. Eliciting and detecting a *reply*

We denote by $\mathcal{L}(I)$ the output generated by a LLM $\mathcal{L}$ when prompted with input $I$. This output is treated as an ordered list of words and invisible characters. Unlike the embedding stage, these extracted invisible characters are not assumed to be in a syllable of size $m$. For a watermarked document $D^w$, the verification consists of constructing $\lfloor \frac{\lceil \Delta/\delta \rceil}{2} \rfloor$ inputs $(ID_i^w)_{i=1..\lfloor \frac{\lceil \Delta/\delta \rceil}{2} \rfloor}$ made of:

- $D^{wc}_{2(i-1)+1}$, containing the repeated $j - t$ syllables of the *cue wc*, followed by

- the first $t \cdot (1 + step)$ elements of $D^{wr}_{2i}$ (i.e. the first $step \cdot t$ words of $D^{wr}_{2i}$ embedded with the tail of the *cue wc*). This completes the partial *cue* of $D^{wc}_{2(i-1)+1}$ and serves as a stimulus for the appearance of the *reply wr*.

We then query the model $\lambda$ times with each $ID_i^w$. The chunk-level verification, described in Algorithm 2 outputs one if and only if the *reply wr* is detected in $\mathcal{L}(ID_w^i)$ in at least one of the $\lambda$ repetitions. Detection is conducted by filtering the model's output, removing all visible characters, and concatenating only the invisible ones in their original order into $o_{\text{invis}}$. If any subsequence of $o_{\text{invis}}$ matches $wr$, the *reply* has been detected.

---

**Algorithm 2** $verif$, searching for *reply* in $\mathcal{L}(ID_i^w)$)

---

**Input: LLM** $\mathcal{L}$**, a prompt** $ID_i^w$**, repetition factor** $\lambda$
**Output: 1 if the *reply* is detected, 0 else**

 1: **for** $x = 1..\lambda$ **do**
 2:    $o \leftarrow \mathcal{L}(ID_i^w)$
 3:    $o_{\text{invis}} \leftarrow$ empty list
 4:    **for** $o_i \in o$ **do**
 5:      **if** $o_i \cap \mathcal{A} \neq \emptyset$ **then**
 6:         **for all** $a \in o_i \cap \mathcal{A}$ in the order they appear in $o_i$ **do**
 7:            Append $a$ to $o_{\text{invis}}$
 8:         **end for**
 9:      **end if**
10:    **end for**
11:    Flatten $wr$ into a single sequence of invisible characters $w_{\text{flat}}^r$
12:    **if** $w_{\text{flat}}^r$ is a contiguous subsequence of $o_{\text{invis}}$ **then**
13:      **return** 1
14:    **end if**
15: **end for**
16: **return** 0

---

## A.4. Ranking test to assess membership

Algorithm 3 describes the ranking test procedure to determine membership. We do not assume that the collection was either wholly used or wholly unused during fine-tuning. Instead, we aim to determine whether elements of the collection have been used.

---

**Algorithm 3** $decision$, assess membership

---

**Input:** $\mathcal{L}, \mathcal{D},$ **rank** $k, \lambda, \boldsymbol{w}, \boldsymbol{W_K}$
**Output:** $\exists d^{\boldsymbol{w}} \in \mathcal{D}^{\boldsymbol{w}}$ **on which** $\mathcal{L}$ **was trained**

1: $s \leftarrow verif(\mathcal{L}, D^w, \lambda)$
2: rank $\leftarrow 1$
3: **for all** $w_K \in W_K$ **do**
4:    $s_K \leftarrow verif(\mathcal{L}, D^{w_K}, \lambda)$
5:    **if** $s_K \geq s$ **then**
6:      rank $\leftarrow$ rank + 1
7:    **end if**
8: **end for**
9: **return** rank $\leq$ k

---

# B. Experiment Details

Table 5 presents the experimental setup, including the low-rank adaptation (LoRA; Hu et al., 2022) configuration, training parameters, and generation settings for each experiment. All experiments were conducted on NVIDIA H100 and H200 GPUs under identical experimental settings.

*Table 5.* LoRA configuration and training setup.

| Parameter | Value |
|---|---|
| **LoRA configuration** | |
| Target modules | q_proj, k_proj, v_proj, o_proj |
| Rank ($r$) | 12 |
| Alpha | 32 |
| Dropout | 0.05 |
| Bias | none |
| Task type | Causal LM |
| **Training setup** | |
| Base model | Mistral-7B-v0.1, LLaMA-2-7B-hf and GPT-OSS-20B |
| Dataset split | 90% train / 10% eval |
| Sequence length | 4096 tokens |
| Attention | FlashAttention 2 (Dao, 2023) for Mistral and LLaMA and eager for GPT-OSS |
| Precision | bfloat16 for Mistral and LLaMA and MXFP4 for GPT-OSS |
| Optimizer | paged_adamw_8bit (Loshchilov & Hutter, 2019; Dettmers et al., 2022) |
| Learning rate | $2 \times 10^{-4}$ (cosine schedule) |
| Warmup ratio | 0.03 |
| Weight decay | 0.05 |
| Batch size | 2 per device |
| Grad. accumulation | 8 steps (effective batch size 16) |
| Epochs | 3 |
| **Generation configuration** | |
| Samlpe | True |
| Temperature | 0.7 |
| Top p | 0.9 |
| Top k | 50 |
| Max new tokens | 200 |

# C. Watermarked Text Statistics

Table 6 summarizes basic length statistics of the watermarked documents used for finetuning in Sections 4.2 and 4.3. For each dataset and each configuration, we report the number of watermarked documents as well as the minimum, maximum, mean, and standard deviation of document length measured in number of words.

In the experiments of Section 4.3, only the size of the overall training set is varied, while the set of watermarked documents is kept fixed. Consequently, all configurations in that setting share identical watermarked texts and therefore identical statistics in Table 6.

*Table 6.* Statistics of watermarked documents in the **Poems**, **Blog**, and **News** datasets. The last column approximates the number of watermark repetitions per document, assuming one occurence per 32 words.

| Dataset | #WMed Docs | Doc. Length | | | | ≈ WM rep./Doc |
| --- | --- | --- | --- | --- | --- | --- |
| | | Min | Max | Mean | Std | |
| **Poems** | | | | | | |
| | 10 | 214 | 532 | 336.0 | 102.0 | 10 |
| | 20 | 203 | 592 | 342.9 | 105.4 | 11 |
| | 30 | 203 | 592 | 331.0 | 104.5 | 10 |
| | 40 | 202 | 809 | 325.0 | 128.0 | 10 |
| | 50 | 202 | 914 | 333.9 | 147.0 | 10 |
| | 60 | 202 | 914 | 334.8 | 140.9 | 10 |
| | 70 | 202 | 914 | 339.4 | 136.1 | 11 |
| | 80 | 202 | 914 | 338.9 | 145.7 | 11 |
| | 90 | 202 | 914 | 333.3 | 141.8 | 10 |
| | 100 | 202 | 1996 | 349.1 | 222.2 | 11 |
| **Blog** | | | | | | |
| | 10 | 252 | 934 | 417.9 | 181.0 | 13 |
| | 20 | 222 | 934 | 383.1 | 151.5 | 12 |
| | 30 | 215 | 1264 | 400.7 | 213.2 | 13 |
| | 40 | 204 | 1264 | 377.3 | 194.7 | 12 |
| | 50 | 204 | 1264 | 387.2 | 200.5 | 12 |
| | 60 | 204 | 1264 | 375.1 | 188.0 | 12 |
| | 70 | 203 | 1264 | 361.9 | 179.6 | 11 |
| | 80 | 203 | 1836 | 370.5 | 238.5 | 12 |
| | 90 | 203 | 1836 | 361.6 | 228.0 | 11 |
| | 100 | 203 | 1836 | 357.3 | 219.3 | 11 |
| **News** | | | | | | |
| | 1 | 1665 | 1665 | 1665.0 | 0.0 | 52 |
| | 5 | 1427 | 1743 | 1570.0 | 119.2 | 49 |
| | 10 | 1369 | 1743 | 1506.4 | 118.2 | 47 |
| | 20 | 1329 | 1743 | 1508.0 | 134.8 | 47 |
| | 30 | 1270 | 1743 | 1492.7 | 124.9 | 47 |
| | 40 | 1240 | 1743 | 1471.7 | 126.3 | 46 |
| | 50 | 1240 | 1743 | 1475.8 | 125.8 | 46 |
| | 60 | 1240 | 1743 | 1484.0 | 122.2 | 46 |
| | 70 | 1240 | 1743 | 1479.6 | 122.0 | 46 |
| | 80 | 1240 | 1743 | 1484.1 | 123.0 | 46 |
| | 90 | 1240 | 1743 | 1480.5 | 120.6 | 46 |
| | 100 | 1240 | 1743 | 1475.3 | 120.7 | 46 |

Table 7 presents the statistics of the watermarked texts used in finetuning in Section 4.4 as well as in the additional experiment conducted using GPT-OSS. In these experiments, the number of documents associated with each unique watermark was kept fixed, while only the number of unique watermarks ($U$) was varied. GPT-OSS uses a larger number of documents per unique watermark (100) to account for its increased model size.

*Table 7.* Statistics of watermarked documents with varying numbers of unique watermarks ($U$). The last column approximates the number of watermarks per document, assuming one watermark per 32 words.

| Data | Model | $U$ | Doc/WM | Doc. Length | | | $\approx$ WM rep./Doc |
|------|-------|-----|--------|-----|-----|------|-----------------|
| | | | | Min | Max | Mean | |
| **Blog** | LLaMA | 5 | 40 | 200 | 1357 | 352.8 | 11 |
| | LLaMA | 20 | 40 | 200 | 2743 | 368.3 | 12 |
| | LLaMA | 30 | 40 | 200 | 2743 | 364.0 | 11 |
| | LLaMA | 40 | 40 | 200 | 2743 | 361.5 | 11 |
| | LLaMA | 50 | 40 | 200 | 2743 | 359.9 | 11 |
| | Mistral | 5 | 60 | 200 | 1357 | 359.1 | 11 |
| | Mistral | 20 | 60 | 200 | 2743 | 363.3 | 11 |
| | Mistral | 30 | 60 | 200 | 2743 | 361.5 | 11 |
| | Mistral | 40 | 60 | 200 | 2743 | 360.1 | 11 |
| | Mistral | 50 | 60 | 200 | 2743 | 357.5 | 11 |
| | GPT-OSS | 5 | 100 | 200 | 1836 | 354.6 | 11 |
| | GPT-OSS | 20 | 100 | 200 | 2743 | 357.8 | 11 |
| | GPT-OSS | 30 | 100 | 200 | 2743 | 358.8 | 11 |
| | GPT-OSS | 40 | 100 | 200 | 2743 | 358.4 | 11 |
| | GPT-OSS | 50 | 100 | 200 | 2743 | 356.3 | 11 |
| **Poems** | Mistral/LLaMA | 5 | 40 | 200 | 3172 | 424.4 | 13 |
| | Mistral/LLaMA | 20 | 40 | 200 | 9722 | 450.1 | 14 |
| | Mistral/LLaMA | 30 | 40 | 200 | 9722 | 454.9 | 14 |
| | Mistral/LLaMA | 40 | 40 | 200 | 9722 | 499.1 | 16 |
| | Mistral/LLaMA | 50 | 40 | 200 | 9722 | 491.6 | 15 |
| | GPT-OSS | 5 | 100 | 200 | 8391 | 525.5 | 16 |
| | GPT-OSS | 20 | 100 | 200 | 9722 | 460.3 | 14 |
| | GPT-OSS | 30 | 100 | 200 | 9722 | 460.3 | 14 |
| | GPT-OSS | 40 | 100 | 200 | 9722 | 484.9 | 15 |
| | GPT-OSS | 50 | 100 | 200 | 9722 | 469.1 | 15 |
| **News** | Mistral/LLaMA | 20 | 40 | 1218 | 1803 | 1486.8 | 46 |
| | Mistral/LLaMA | 30 | 40 | 1218 | 1803 | 1484.2 | 46 |
| | Mistral/LLaMA | 40 | 40 | 1218 | 1803 | 1488.5 | 47 |
| | Mistral/LLaMA | 50 | 40 | 1218 | 1836 | 1500.0 | 47 |
| | GPT-OSS | 5 | 100 | 1240 | 1754 | 1512.2 | 47 |
| | GPT-OSS | 20 | 100 | 1218 | 1803 | 1483.6 | 46 |
| | GPT-OSS | 30 | 100 | 1218 | 1803 | 1479.1 | 46 |
| | GPT-OSS | 40 | 100 | 1218 | 1833 | 1485.8 | 46 |
| | GPT-OSS | 50 | 100 | 881 | 1836 | 1497.1 | 47 |

## D. Per-watermark regurgitation rate in multi-watermark settings

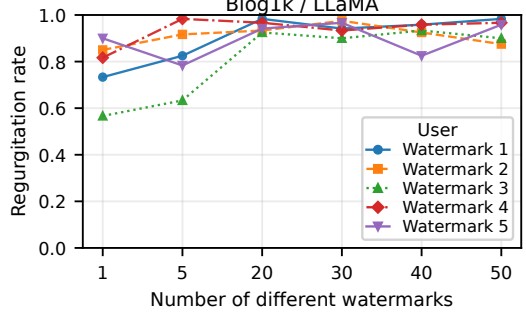
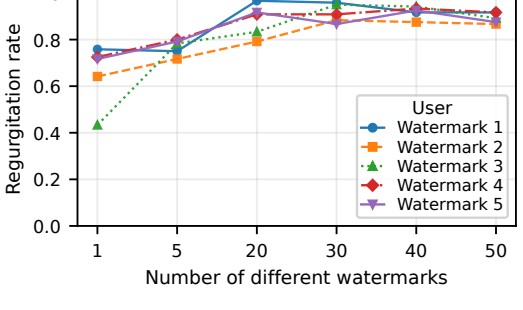

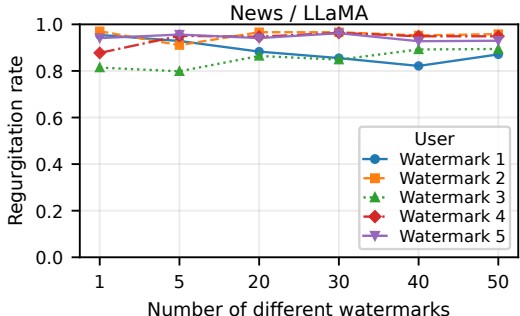
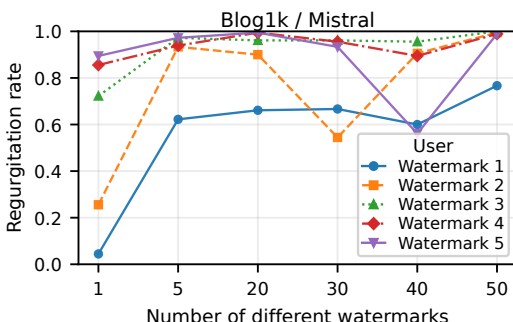

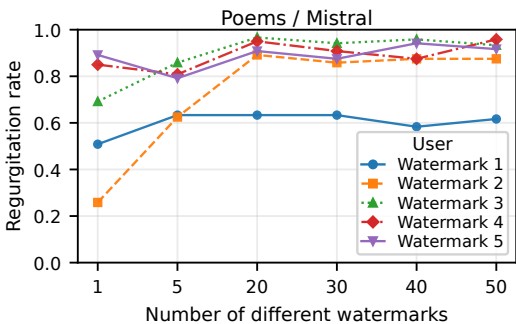
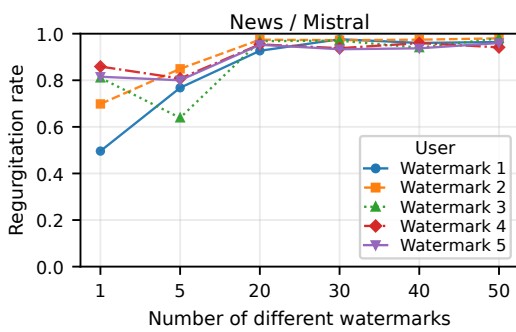

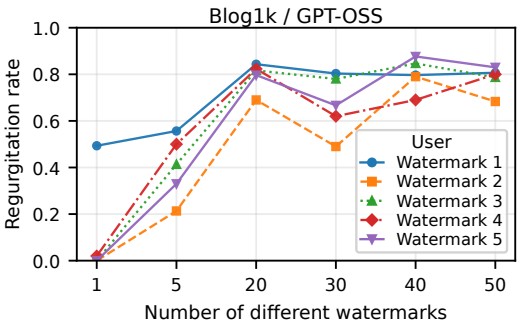
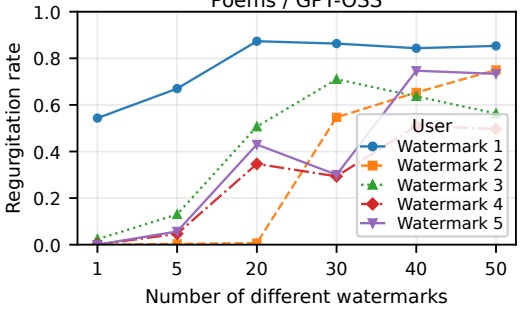

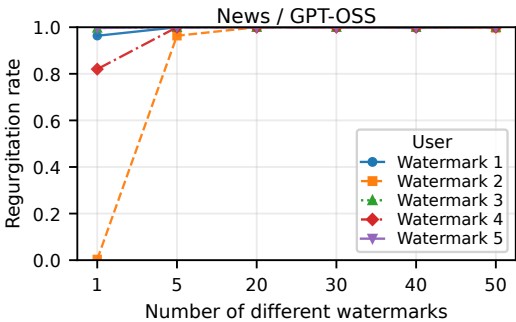

*Figure 6.* Regurgitation rate depending on the number of unique watermarks. Curves show the details of the regurgitation rate across 5 watermarks.

Figure 6 reports the detailed per-watermark results corresponding to Figure 4. It additionally shows results using the same setting for a larger model, GPT-OSS, confirming the general trend. Each subfigure corresponds to a single experimental setup (one model and one dataset), and shows the regurgitation rate of the five individual watermarks tested throughout our experiments as a function of the number of unique watermarks present in the training set. For a given watermark, we compute the regurgitation rate by prompting the model three times for each chunk in the corresponding collection, averaging the chunk-level regurgitation across prompts, and then averaging the resulting scores across all chunks.

Variability among watermarks is important in certain settings, in particular for Mistral. On Blog1k, two watermarks exhibit significantly low regurgitation rates when isolated (0-0.23). As the number of unique watermarks increases, the regurgitation rate of each individual watermark generally improves and eventually reaches a plateau close to perfect regurgitation. A notable exception is watermark 1 for Mistral fine-tuned on Blog1k and Poems, which saturates at approximately 0.6. These outliers account for the relatively large confidence intervals observed when results are aggregated across watermarks.

While some configurations display non-monotonic behavior as the number of watermarks grows (e.g., watermark 2 for Mistral/Blog1k dropping for $U = 30$), this variability may be attributable to the relatively small fine-tuning datasets, which may amplify sensitivity to stochastic effects such as data sampling and optimization noise. Crucially, these fluctuations do not alter the overall trend: increasing the number of unique watermarks in the training data generally leads to improved regurgitation performance.

GPT-OSS exhibits similar qualitative behavior, with lower regurgitation at small $U$ but a steady, consistent increase as $U$ grows, eventually reaching high regurgitation levels across all datasets. However, it shows more uniform behavior across watermarks, with no persistent outliers at large $U$, and variability across watermarks becomes negligible as the number of unique watermarks increases.

# E. Robustness to non-adversarial transformations

Invisible characters have previously been exploited in jailbreaking attacks and are often either unsupported or semantically irrelevant. Consequently, it is plausible that such characters may be ignored or removed by web services or LLM providers. In this section, we examine the extent to which invisible characters and the associated watermarking scheme are preserved under passive transformations, that is, under data-cleaning operations not deliberately designed to be adversarial.

We formulate the following hypotheses, which together span the full lifecycle of data and model interaction from dataset collection and preprocessing to tokenization and post-training detection:

**H1** *Multi-media:* Invisible characters can be embedded across multiple media (e.g., webpages, plain text, and PDF documents) and remain intact following format conversions, ensuring their presence during data collection.

**H2** *Data preparation:* Common data cleaning and preprocessing pipelines for dataset construction do not systematically remove invisible characters or their combination, ensuring their presence in training data.

**H3** *Tokenization:* Invisible characters are correctly identified by standard tokenizers and represented by a consistent and limited number of tokens, allowing for their reliable propagation through model inputs.

**H4** *Public interfaces:* Standard inference endpoints do not systematically filter invisible characters either before or after inference, ensuring that downstream detection and challenge mechanisms remain operational.

These hypotheses are experimentally confirmed hereafter using a vast variety of invisible characters.

## E.1. Alphabet selection

To identify the invisible characters used in our experiments, we tested the 235 Unicode code points belonging to the "Control" and "Format" categories, as these are the most likely to be visually imperceptible. Each character was rendered in a terminal, a text document, and a CSV file. Characters that produced a visible effect, either by displaying a glyph (e.g., U+0600 "Arabic Number Sign") or by altering text layout (e.g., U+202E "Right-to-Left Override"), were excluded. This filtering process yielded a set of 130 invisible characters whose robustness is investigated hereafter.

We emphasize that **invisibility may be context-dependent**, and therefore do not claim that these characters are universally invisible across all contexts and environments. In particular, some characters may affect rendering depending on context (e.g., U+202D, "Left-to-Right Override," in non–left-to-right settings), while others may generate visible glyphs or spacing in certain browsers or text editors. We further exclude the 9 remaining bi-directional characters from the alphabet used in Section 4 [6].

## E.2. Multi-media

Although invisible characters are expected to be consistently embedded in standard text formats such as TXT, JSON, or HTML, we sought to verify their robustness in PDF documents and blog posts or articles. Specifically, we assessed whether these characters are correctly rendered and preserved through content extraction techniques, rather than being removed or altered by servers or conventional web page rendering processes.

**PDF.** A PDF was generated using the ReportLab Python library, and text was subsequently extracted using PyMuPDF. Of the 130 characters, only 20 were successfully recovered while remaining invisible in the PDF. We note that this behavior is font-dependent: using *DejaVuSans*, 20 characters were preserved, whereas *Helvetica* embedded only 2.

We further verified invisibility and recoverability through copy-and-paste using common PDF viewers. Evince and Gmail's integrated viewer recovered all 20 characters, Chrome recovered 19, Acrobat Reader 6, and Firefox recovered none.

---

[6]A complete list and description of the selected characters is available at https://github.com/liam-0/Data-Provenance-Auditing-of-Fine-Tuned-Large-Language-Models-with-a-Text-Preserving-Technique/tree/main/robustness/CharSelection.

**Web.** We evaluated four websites commonly used for dataset construction and AI training: LinkedIn, Reddit, Wikipedia, and GitHub. Invisible characters were inserted into posts, articles, or documents and subsequently recovered via two methods: direct copy-and-paste and inspection of the page source code, simulating web scraping. All 130 characters were successfully recovered on all platforms and, as expected, appeared in the corresponding pages' source code. On Reddit and LinkedIn, some characters (130 and 105, respectively) were returned in a defanged form, i.e., automatically converted or escaped. Despite this conversion, the information encoded by the invisible characters remained fully recoverable.

**Takeaway:** Embedding invisible characters in PDFs is both font-dependent and viewer-dependent: using widely adopted Python libraries and PDF viewers, up to 20 characters could be reliably recovered. In contrast, on the web, all tested platforms preserved all 130 characters, although some were returned in defanged form in the page source.

### E.3. Data preparation pipelines

To evaluate the robustness of our watermarking method, we analyzed and reproduced a series of representative data preprocessing pipelines commonly used in LLM training. Since the exact preprocessing procedures adopted in industry-scale training remain undisclosed, we referred to publicly available open-source datasets and their associated repositories, including C4 (Raffel et al., 2020), CCNet (Wenzek et al., 2020), the Pile (Gao et al., 2020), RedPajama (Weber et al., 2024), Dolma (Soldaini et al., 2024), and FineWeb (Penedo et al., 2024).

We reconstructed the preprocessing workflows as faithfully as possible by reusing official source code when possible, adaptating it with minimal modification when not reusable as-is, and implementing missing functions following methodological details described in the respective papers.

We then tested our watermarking scheme using the same dataset employed in the main experiments. From this dataset, we extracted a representative sentence and replaced either one type of invisible character or all invisible characters with each of the 130 invisible characters described earlier. The modified samples were then processed through the reconstructed preprocessing pipelines. The details of each data-cleaning pipeline are described on git [7].

**Takeaway:** The results in Table 8 indicate that all but one of the evaluated data-cleaning pipelines fail to remove invisible characters. Only the Pile data-cleaning pipeline filters out seven such characters, i.e. $U+206A$–$U+206F$ and $U+FEFF$. We further observe that **no pipeline removes marks**. In some pipelines, text containing invisible characters may receive lower quality scores and therefore be excluded during the quality-filtering stage. Importantly, this exclusion does not constitute removal of the marks themselves; rather, it prevents the affected texts from entering the final dataset. We argue that such behavior aligns with users' interests and helps mitigate unauthorized or illegitimate data usage during deduplication or noise-filtering stages.

Consequently, our watermarking scheme is robust to the realistic preprocessing operations used in large-scale LLM training: watermarks remain intact throughout the reproduced pipelines, provided they do not rely on the seven characters filtered out by the Pile.

*Table 8.* Number of invisible characters removed by each data-cleaning pipeline. No mark is removed otherwise.

| Data Cleaning Pipeline | # filtered Invisible Characters |
|---|---|
| C4 (Raffel et al., 2020) | 0 |
| CCNet (Wenzek et al., 2020) | 0 |
| the Pile (Gao et al., 2020) | 7 |
| RedPajama (Weber et al., 2024) | 0 |
| Dolma (Soldaini et al., 2024) | 0 |
| FineWeb (Penedo et al., 2024) | 0 |

---

[7]See https://github.com/liam-0/Data-Provenance-Auditing-of-Fine-Tuned-Large-Language-Models-with-a-Text-Preserving-Technique/blob/main/TestPipeline/Description%20and%20Technical%20Details%20of%20the%20Data%20Preparation%20Pipeline%20Experiments.pdf for the full description of the data-cleaning pipelines.

### E.4. Tokenizer

We evaluated 10 tokenizers, grouped by their underlying tokenization algorithms:

- **SentencePiece (Unigram)**: *google-t5/t5-small* [8] (Raffel et al., 2020), *google/flan-t5-small* [9] (Chung et al., 2022).

- **OpenAI tokenizers (BPE)**: the official *cl100k_base* tokenizer used by *gpt-3.5-turbo* (Brown et al., 2020) and *gpt-4* (OpenAI, 2024b), and the *o200k_base* tokenizer used by *gpt-4o* (OpenAI, 2024a).

- **Byte-level BPE (GPT-style)**: *FacebookAI/roberta-base* [10] (Liu et al., 2020), *EleutherAI/gpt-neo-125m* [11] (Black et al., 2021), *facebook/opt-125m* [12] (Zhang et al., 2022), *mistralai/Mistral-7B-v0.1* [13] (Jiang et al., 2023), *openlm-research/open_llama_7b* [14] (Geng & Liu, 2023), and *deepseek-ai/DeepSeek-R1* [15] (Guo et al., 2025).

**Tokenization of invisible characters.** The evaluation proceeded in two steps. First, each character was tested in isolation to confirm that it was mapped to at least one token and therefore not ignored by the tokenizer. Second, given the context-dependent nature of tokenization and the fact that invisible characters are inserted at the end of space-separated words in the watermarking scheme, the difference in token count was measured between the string "A " and a string in which an invisible character was inserted between "A" and the following space. The results are summarized in Table 9.

*Table 9.* Distribution of invisible characters by number of additional tokens produced across tokenizers. N/A denotes characters ignored by the tokenizer.

| tokenizer | # Additional token produced | | | | | |
|---|---|---|---|---|---|---|
| | N/A | 0 | 1 | 2 | 3 | 4 |
| *roberta-base* | 0 | 0 | 2 | 21 | 12 | 95 |
| *facebook/opt-125m* | 0 | 0 | 2 | 21 | 12 | 95 |
| *t5-small* | 6 | 0 | 124 | 0 | 0 | 0 |
| *flan-t5-small* | 0 | 6 | 124 | 0 | 0 | 0 |
| *gpt-neo-125M* | 0 | 0 | 2 | 21 | 12 | 95 |
| *Mistral-7B-v0.1* | 0 | 0 | 11 | 1 | 13 | 105 |
| *open_llama_7b* | 0 | 0 | 1 | 1 | 23 | 105 |
| *DeepSeek-R1* | 0 | 0 | 5 | 20 | 9 | 96 |
| *gpt-3.5-turbo & gpt-4* | 0 | 0 | 4 | 20 | 106 | 0 |
| *gpt-4o* | 0 | 0 | 11 | 13 | 42 | 64 |

Only t5-small ignored six characters. Notably, flan-t5-small recognized all invisible characters, and for six of them, the insertion did not increase the token count, indicating that these characters were merged with the preceding "A" into a single token.

**Comparison with innocuous, semantic-carrying symbols.** Another point of analysis is if the resulting number of tokens is reasonable. To assess this, we compared their tokenization behavior with that of the 1,376 Unicode characters classified as "Emoji." Emojis provide a relevant point of comparison because, unlike invisible characters, they are widely used, semantically meaningful symbols that appear legitimately in natural text and are fully supported by modern tokenizers.

Figure 7 reports the distribution of additional tokens produced when inserting either a random invisible character or a random emoji, aggregated across all remaining tokenizers. Both character types exhibit a similar probability of producing exactly one additional token (approximately 22% for invisible characters and 21% for emojis). However, their higher-order behavior differs: emojis more frequently result in intermediate token counts (2–3 tokens), whereas invisible characters are more likely to trigger extreme outcomes, including generating four tokens or, in some cases, no additional token.

---

[8] https://huggingface.co/google-t5/t5-small
[9] https://huggingface.co/google/flan-t5-base
[10] https://huggingface.co/FacebookAI/roberta-base
[11] https://huggingface.co/EleutherAI/gpt-neo-125m
[12] https://huggingface.co/facebook/opt-125m
[13] https://huggingface.co/mistralai/Mistral-7B-v0.1
[14] https://huggingface.co/openlm-research/open_llama_7b
[15] https://huggingface.co/deepseek-ai/DeepSeek-R1

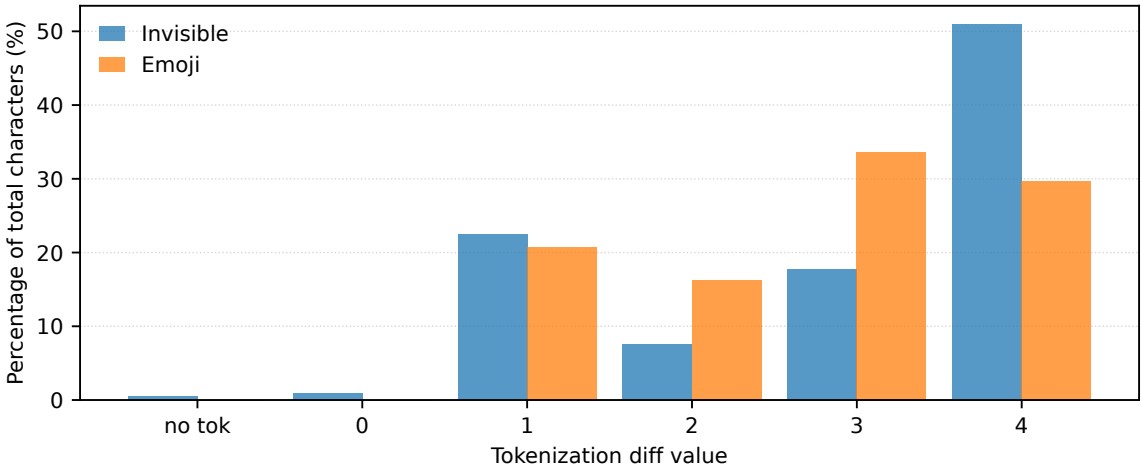

*Figure 7.* Number of token distribution: invisible characters vs emojis.

**Takeaway:** Tokenizers consistently recognize invisible characters, demonstrating substantial consistency in preserving these characters. Interestingly, despite being frequent, visually salient, and semantically meaningful, emojis are not tokenized more effectively than invisible characters. In fact, invisible characters are more likely to yield a low ($\leq 1$) number of additional tokens, highlighting their subtlety and the potential for low-visibility inputs to be incorporated reliably into token sequences.

### E.5. Public interfaces

We aim to assess whether the public interfaces of widely used models filter out invisible characters, for example, as a security mechanism to prevent prompt injection or jailbreaking. To this end, we used the free public GUIs of chatbots. This approach is consistent with evaluating fine-tuned chatbots that may not expose a public API, and it reflects the common assumption that public GUIs are generally more restrictive than APIs. For instance, ChatGPT's web interface implements prompt isolation and moderation techniques that are either optional or less strict when accessing the model via the API.

The tested characters were inserted between two visible characters, "A" and "B". The resulting text was submitted to the free public graphical interfaces of three of the most commonly used free chatbots[16], Deepseek[17], Le Chat[18], and ChatGPT[19], as direct text input and as a plain text (.txt) document with an innocuous filename. In each case, the model was instructed to repeat the input verbatim. Links to the corresponding chat sessions, prompts, and outputs are available on git.[20]

Results were the following:

- **DeepSeek** successfully reproduced all 130 invisible characters in both input modes.

- **Le Chat** reproduced all 130 characters when provided as direct text input. However, when given the text as a document, it returned the Unicode escape sequences of the characters rather than the characters themselves.

- **ChatGPT** was the only API in which filtering occurred: it reproduced only 32 of the invisible characters when the text was provided as direct input, preserving one additional character for documents input.

**Takeaway:** All-but-one tested graphical interfaces do not filter invisible characters, either through prompt sanitization or

---

[16]see e.g. https://www.visualcapitalist.com/the-10-most-used-ai-chatbots-in-2025
[17]https://chat.deepseek.com/
[18]https://chat.mistral.ai/chat
[19]https://chatgpt.com
[20]See https://github.com/liam-0/Data-Provenance-Auditing-of-Fine-Tuned-Large-Language-Models-with-a-Text-Preserving-Technique/blob/main/robustness/TestsAPI/README.md for the corresponding chat sessions, prompts, and outputs.

posterior to inference. Only ChatGPT failed to reproduce the full set of characters, although it still preserved a substantial subset of 33 invisible symbols, sufficient to constitute a non-trivial encoding alphabet.

### E.6. Summary

Table 10 summarizes how many invisible characters are preserved across the tests described above. Overall, these characters are only mildly affected, with the main exceptions occurring when they are defanged in source code or embedded within PDFs.

*Table 10.* Summary of invisible character preservation across different interfaces and processing pipelines. The total number of inserted invisible characters is 130.

| Category | Processing Component | # Preserved Characters |
|---|---|---|
| PDF | Script extraction | 20 |
| | Copy/paste (Evince) | 20 |
| | Copy/paste (Gmail) | 20 |
| | Copy/paste (Chrome) | 19 |
| | Copy/paste (Acrobat) | 6 |
| | Copy/paste (Firefox) | 0 |
| Web | Git | 130 |
| | Reddit | 130 |
| | Reddit (source) | 0 (130 defanged) |
| | LinkedIn | 130 |
| | LinkedIn (source) | 25 (+105 defanged) |
| | Wikipedia | 130 |
| | Wikipedia (source) | 130 |
| Chatbot GUI | Le Chat | 130 |
| | Le Chat (.txt) | 0 (130 defanged) |
| | DeepSeek | 130 |
| | DeepSeek (.txt) | 130 |
| | ChatGPT | 32 |
| | ChatGPT (.txt) | 33 |
| Tokenizer | T5-small | 124 |
| | All others | 130 |
| Data Cleaning Pipeline | C4 | 130 |
| | CCNet | 130 |
| | The Pile | 123 |
| | RedPajama | 130 |
| | Dolma | 130 |
| | FineWeb | 130 |

## F. Robustness to untargeted, generic watermark removal attacks

While active adversaries are not considered in our threat model, we nonetheless evaluate the robustness of our scheme with comparison to the only state-of-the-art text-preserving watermarking technique, the homoglyphs setting of Wei et al. (2024). We use the generic adversarial modifications proposed in Piet et al. (2025), which were originally designed to assess the robustness of watermarking methods for model outputs. They generally rewrite a watermarked text, aiming at preserving its semantic while removing watermarks, without knowledge of the watermarking scheme. More specifically, we evaluate the following attacks:

- *Lowercase* converts alphabetic characters to lowercase.

- *Typo* introduces random character-level spelling perturbations with probability $p = 0.1$.

- *Synonym* replaces words with synonyms with probability $p = 0.5$.

- *Adjacent-word swap* swaps neighboring words and therefore changes local word order with probability $p = 0.1$.

- *Repeat/Delete* randomly repeats or deletes tokens, producing stronger local corruption with probability $p = 0.05$.

- *Paraphrase* rewrites the text with a kalpeshk2011/dipper-paraphraser-xxl paraphrasing model (Krishna et al., 2023) [21].

- *Back-translation* uses Argos Translate[22] to translate the text from English into French or Russian and then back into English, often changing both wording and sentence structure.

Table 11 reports the average per-document percentage of watermarks that remain preserved after each transformation, computed over 2,000 documents, for both our method and Homogplhys (Wei et al., 2024).

*Table 11.* Robustness under untargeted, generic removal attacks. Values report the percentage of preserved detectable watermarks.

| Attack | Ours (%) | Homogplyphs (Wei et al., 2024) (%) |
|---|---|---|
| Lowercase | 100 | 0 |
| Typo ($p = 0.1$) | 100 | 0 |
| Synonym ($p = 0.5$) | 99.9 | 90.7 |
| Adjacent-word swap ($p = 0.1$) | 99.7 | 1.2 |
| Repeat/Delete ($p = 0.05$) | 58.4 | 0 |
| Paraphrase | 0 | 0 |
| Back-translation | 0 | 0 |

The results show that our watermarking scheme is robust to several low-cost, semantics-preserving perturbations and is substantially more stable than the baseline under these transformations. However, our method is highly vulnerable to paraphrasing and back-translation, the two most costly attacks, as they require auxiliary models. Applying such transformations to an entire training dataset would demand substantial computational resources while also degrading data quality.

---

[21]https://huggingface.co/kalpeshk2011/dipper-paraphraser-xxl
[22]https://www.argosopentech.com/

