# OpenReview forum: "Data Provenance Auditing of Fine-Tuned Large Language Models with a Text-Preserving Technique"
_ICML.cc/2026/Conference — ICML 2026 regular_

### Official Review · Reviewer_qCNk · 2026-03-10

**Soundness:** 3
**Presentation:** 3
**Significance:** 3
**Originality:** 3
**Overall Recommendation:** 3
**Confidence:** 4

**Summary:**

The paper proposes a text-preserving watermarking method by inserting invisible Unicode characters. It introduces a "cue,reply" trigger structure and detects the watermark using a counterfactual watermark ranking test. Compared with simple threshold-based detection, the ranking-based statistical test provides a more principled decision rule.

**Compliance With Llm Reviewing Policy:**

Affirmed.

**Key Questions For Authors:**

N/A

**Limitations:**

See weaknesses.

**Strengths And Weaknesses:**

### Strengths

The paper presents a carefully designed framework for black-box dataset provenance auditing. The combination of invisible watermarking, trigger-based memorization, and ranking-based statistical testing is combined for auditing whether a dataset was used during fine-tuning.

### Weaknesses


- The robustness of the watermark against removal is limited, which may hinder its use in practical systems. The method relies on invisible Unicode characters, which can be easily removed by common preprocessing steps in real data pipelines, such as Unicode normalization or re-tokenization. If the training data undergoes such preprocessing, the watermark may disappear entirely. Moreover, it should be easy to observe that there are some repeated Unicode patter in several corpus. Then, if the model provider is aware of the watermark, he/she could remove suspicious Unicode patterns or filter repeated characters. It is also unclear whether an attacker could inject additional watermark patterns into the same document to dilute the learnability of the original watermark. More fundamentally, the approach effectively trains the model to learn a trigger–response pattern (cue, reply). The success of detection therefore relies on the model memorizing this pattern. However, modern training practices such as strong regularization, RLHF, or alignment training may suppress such memorization and reduce detection reliability.

- There could be potential watermark forgery. If an adversary can insert watermark patterns into arbitrary datasets, it may become possible to fabricate provenance evidence or confuse the attribution process. The authors should clarify the threat model and the mechanism that prevents watermark forgery.

---

> ### Author Rebuttal · Authors · 2026-03-31
>
> We thank the reviewer for the thoughtful evaluation. We address three concerns: robustness to removal, dependence on memorization, and watermark forgery. We clarify that our method is robust to realistic preprocessing and many low-cost perturbations, substantially more robust than prior work under such transformations. We agree that more costly adversarial attacks (e.g., paraphrasing) can remove the watermark.
>
> **1. Robustness to preprocessing vs. active removal**
>
> Our primary target setting is dataset provenance auditing under non-adversarial preprocessing, while active removal is out of scope.
>
> In non-adversarial preprocessing, we already provide evidence (Appendix E.3) that our watermark survives realistic pipelines: 5 out of 6 common preprocessing pipelines preserve all characters, and the remaining one removes only a small subset. The same applies to tokenization. We show (Appendix E.4) that the tested tokenizers do not simply ignore the proposed characters in general, but typically preserve them as tokenizable signal. In practice, robustness can also be further improved by restricting the alphabet to pipeline-safe characters while retaining a large watermark space (Appendix A.1). Hence, under ordinary collection, cleaning, and tokenization workflows, the watermark remains practically stable.
>
> We agree that an active provider aware of the defense may attempt to normalize away zero-width characters, filter suspicious /repeated Unicode, or paraphrase documents before training. We already acknowledge this limitation in Section 6.
>
> To further address this concern, we evaluate standard adversarial transformations from MARKMyWORDS (SaTML 2025) and compare to Wei et al. (ACL Findings 2024):
>
> | Attack                  | Ours (%) | Wei et al. (2024) (%) |
> |------------------------|----------|------------------------|
> | Lowercase              | 100      | 0                      |
> | Typo (p=0.1)           | 100      | 0                      |
> | Synonym (p=0.5)        | 99.9     | 90.7                   |
> | Adj-word swap          | 99.7     | 1.2                    |
> | Repeat/Delete          | 58.4     | 0                      |
> | Paraphrase             | 0        | 0                      |
> | Back-translation       | 0        | 0                      |
>
> These results show that our method is robust to a broad class of low-cost, semantics-preserving perturbations, and substantially more robust than prior work. The only fully successful attacks are paraphrasing and back-translation, which require external models and induce substantial rewriting of the text before training. Importantly, such attacks correspond to active adversaries and fall outside the non-adversarial preprocessing setting targeted by our work.
>
> More generally, such attacks affect completeness rather than soundness: they may reduce detection power (increase false negatives), but do not compromise the false-positive control of the ranking-based test.
>
> **2. Memorization and training regime**
>
> We agree that our method relies on learning a cue–response association. This is by design: memorization enables black-box auditing. Our empirical results show that this signal is reliably learned in supervised fine-tuning (including parameter-efficient settings), even with limited watermark exposure.
>
> We do not claim robustness across all training regimes. In particular, strong regularization or alignment procedures (e.g., RLHF) may attenuate the signal. We will clarify this explicitly in the main text so that the scope is unambiguous, in Section 6, where we already mention the limitation related to RLHF.
>
> **3. Watermark forgery and threat model**
>
> Our setting follows a passive (honest-but-curious) adversary, consistent with prior work on membership inference. The auditor tests whether a dataset was used for training, without assuming that the model provider actively modifies the data to evade detection.
>
> Importantly, our protocol already mitigates a key form of forgery: ex post cherry-picking. Watermarks are sampled from a candidate set and committed at assignment time, while unused candidates serve as counterfactuals in the ranking test. This prevents constructing a watermark after observing the model.
>
> A stronger attack remains possible if an adversary deliberately reuses or re-injects a known watermark into unrelated data (contamination/replay). We do not claim to fully solve this stronger setting in the current paper, and we will clarify it as a limitation in the revision.

---

> > ### Author Rebuttal · Reviewer_qCNk · 2026-04-03
> >
> > Thank you for the detailed response, also adding the experiments. I remain concerned about practicality. Simple cleaning steps (e.g., removing suspicious Unicode or filtering repeated characters) can eliminate the watermark at negligible cost. While Appendix E.3 shows robustness under non-adversarial pipelines, even mildly aware preprocessing would likely invalidate the method. The drop to 58.4% under “Repeat/Delete” already suggests significant fragility. Moreover, even without adversarial intent, practitioners would likely remove unusual or repeated Unicode patterns during routine cleaning, which would naturally eliminate the watermark.

---

> > > ### Author Response · Authors · 2026-04-06
> > >
> > > We thank the reviewer for their answer and we agree with the central point: simple cleaning steps that remove suspicious or repeated Unicode can substantially weaken, and in some cases eliminate, the watermark signal. This can arise both from deliberate suppression and from stronger routine normalization choices, and we will make this boundary more explicit in the revision.
> > >
> > > A useful distinction here is between routine preprocessing introduced for *standard collection or cleaning purposes*, and filtering steps introduced *specifically to suppress the audit signal*. Our empirical claim is limited to the former: under the 6 real collection, preprocessing, and tokenization pipelines tested in Appendix E, the signal is largely preserved. This supports practicality in a range of realistic workflows, but does not imply robustness to arbitrary cleaning or targeted removal. This also includes settings where no targeted removal step is introduced, such as cooperative or semi-cooperative deployments (e.g., fine-tuning services), where providing auditable evidence of data inclusion is desirable.
> > >
> > > We also agree that the 58.4% result under Repeat/Delete indicates meaningful fragility at the perturbation level. However, this does not directly imply failure of the final audit, because the decision aggregates evidence across many chunks and documents rather than relying on a single intact occurrence. In that sense, removal primarily affects completeness, potentially to zero, but does not undermine the soundness of a positive finding when the signal is present.
> > >
> > > We will revise the paper to sharpen this scope: the contribution is a text-preserving black-box auditing method with explicit false-positive control via counterfactual ranking, not a tamper-proof defense against deliberate watermark removal.

---

### Official Review · Reviewer_JehZ · 2026-03-11

**Soundness:** 4
**Presentation:** 4
**Significance:** 2
**Originality:** 4
**Overall Recommendation:** 5
**Confidence:** 2

**Summary:**

The paper considers the problem of marking text to detect if an LLM was trained on that text. The goal is to design a protocol that can mark text without changing its appearance to honest users, and then if it is used for training, a verification procedure on an LLM trained on that text should succeed with high probability if many marked documents were used, and have a limited false positive rate. The authors propose a scheme based on non-visible unicode characters, i.e. ones that affect the tokenization of an example but not the visible text seen by an honest user. The water marks are sequences of "words" made of invisible characters. The authors split each watermark into a unique cue and reply pair. The watermark words are interleaved with the real text in the document; multiple watermarks can be inserted into a document. To detect a specific watermark, the LLM is prompted using the first part of a chunk that had 1 watermark interleaved, and we counts the occurrences of the reply in the output. A user is given a set of K collections of watermarks for each document they wish to mark, and chooses 1 to use for marking; the other K-1 are used as counterfactuals, and we detect training on the data if the watermark collection used on the data was in the top k most detected. Hence as long as the user chooses their watermarks randomly from the collections, the FPR is at most k/K. The authors run experiments to validate their approach on datasets consisting of blogs, poems, and news. The experiments validate the high true positive rate / low false positive rate of their protocol, especially as the number of watermarked documents increases, and that the protocol is robust to the number of unmarked documents,.

**Compliance With Llm Reviewing Policy:**

Affirmed.

**Final Justification:**

The discussion with other reviewers has raised concerns about simple filtering mechanisms, but I felt at the time of my initial review the authors were being transparent about this limitation and argued that brittleness to adaptive adversaries was standard in the watermarking literature. So the review process has not changed my assessment meaningfully.

**Key Questions For Authors:**

No significant questions for the authors.

**Limitations:**

Yes

**Strengths And Weaknesses:**

Strengths:
* Strong and thorough empirical results validate the approach
* The approach is relatively easy to implement in practice
* The authors do a good job explaining the properties that are desirable for a watermarking scheme and how their scheme satisfies these properties
* The watermarking scheme has some natural desirable properties that clearly separate it from past work, the big one being that it doesn't alter the content being watermarked as presented to honest users
* The approach is novel, i.e. past watermarking approaches which apply to text do not make clever use of invisible characters and these enable the nice properties of the approach.

Weaknesses
* As the authors discuss in their limitations, the main weakness is that the approach is not robust to adaptive adversaries, i.e. a malicious LLM trainer aware of the watermarking scheme could add a preprocessing step to filter out the watermarks from training (but this is shared by all watermarking schemes). To be honest I think this makes it unlikely any individual watermarking scheme is likely to be useful in the long-term as they turn data provenance detection into a "cat and mouse game", but I'm still giving a favorable rating to the paper because this seems inherent to the approach of watermarking which has been one the broader research community has interest in, and it's still a good short-term approach as further solutions develop.

---

> ### Author Rebuttal · Authors · 2026-03-31
>
> We thank reviewer JehZ for the positive assessment. We agree that vulnerability to adaptive removal is a fundamental limitation of text watermarking-based provenance methods, rather than a weakness specific to our approach. In that sense, watermarking should be viewed as a practical mechanism within a broader set of auditing tools.
>
> We nevertheless believe that when not fully robust to adaptive adversaries, watermarking can provide auditable signals of data provenance in black-box settings, where alternative mechanisms are currently limited. In particular, it enables data owners or auditors to obtain evidence of likely data use, rather than guarantees, which is often the relevant requirement in real-world scenarios such as compliance checks or dispute resolution.
>
> More broadly, this perspective is consistent with how technical protection measures are treated in existing legal frameworks. For example, EU law provides protection against the circumvention of such "effective technological measures" suggesting that even imperfect mechanisms can have practical and legal relevance when they contribute to access control or rights management. In this light, watermarking can be seen not as a definitive barrier, but as a signal that increases the cost and risk of misuse, and supports accountability in data usage.
>
> We will revise the paper to improve the positioning of our contribution along these lines: as a practically deployable, black-box auditing mechanism that provides meaningful evidence of data provenance, while acknowledging its limitations under fully adaptive adversaries.

---

> > ### Author Rebuttal · Reviewer_JehZ · 2026-03-31
> >
> > Thanks to the authors for their response. I will follow the discussion with other reviewers, but in the meantime I am happy to keep my score.

---

> > > ### Author Response · Authors · 2026-04-06
> > >
> > > We are glad that the reviewer is confirming that his main concerns have been adequately addressed. In the revision, we will sharpen the paper's positioning as a black-box provenance auditing mechanism with explicit false-positive control, while making its limits under targeted removal more prominent.

---

### Official Review · Reviewer_T7RE · 2026-03-12

**Soundness:** 3
**Presentation:** 3
**Significance:** 3
**Originality:** 3
**Overall Recommendation:** 4
**Confidence:** 4

**Summary:**

This paper studies the problem of black-box dataset provenance auditing for fine-tuned LLMs, under the assumption that the data owner can proactively mark documents before release. It uses a cue/reply structure to embed invisible Unicode-based watermarks into text. Then, by prompting a suspicious model with text chunks containing the cue and checking whether the model verbatim reproduces the corresponding reply, the method performs auditing. In addition, the paper proposes a ranking test over reserved counterfactual watermarks to effectively control the false positive rate.

**Compliance With Llm Reviewing Policy:**

Affirmed.

**Key Questions For Authors:**

This paper addresses the problem of determining, in a black-box setting, whether private data has been used for training, which is a very interesting direction. The proposed method solves this problem by injecting invisible characters into text in a watermark-like manner, and it has a certain degree of novelty. However, I have the following concerns:

1. The writing can be improved. For example, the Abstract could include more quantitative information, and the paper should add a paragraph describing the threat model, i.e., what specific real-world scenario this method is intended for.
2. From a methodological perspective, the approach has some novelty, but in the real world it is very common to apply necessary text filtering (e.g., removing invisible characters). As also emphasized by the authors in the limitations section, this gives the method an inherent and direct weakness that is difficult to avoid.
3. I do not understand the statement “we group them into syllables to improve memorization.” Why would LLMs memorize syllables more easily? Is there any experimental validation for this?
4. From Section 3.3, “A document D is split by spaces...”, it seems that this method would have difficulty handling languages such as Chinese and Japanese that do not use spaces. Does this paper implicitly restrict itself to English? This usage scenario should be clarified.
5. The target models in the experiments are LLaMA-2-7b-hf and Mistral-7B-v0.1, both of which are somewhat outdated, and the fine-tuning datasets used in the experiments are also relatively small. Why is the result for nbdocs = 50 on the News dataset missing?

Overall, I think the figures and organization of the paper are good, but some details still need improvement. The motivation of the paper is strong, but the method has substantial limitations.

Minors:

1. $D \in \mathcal{D}$ could be changed to use a lowercase letter; the current notation is not easy to read.
2. The caption of Figure 1 is too brief and does not end with a period.

**Limitations:**

See above.

**Strengths And Weaknesses:**

Strengths:

- Clear motivation
- Important problem
- Well organized

Weaknesses:

- Limited robustness analysis
- The method has limitations
- The experimental models are outdated

---

> ### Author Rebuttal · Authors · 2026-03-31
>
> We thank Reviewer T7RE for the constructive feedback and helpful suggestions.
>
> > 1. [a] The writing can be improved. For example, the Abstract could include more quantitative information, [b] the paper should add a paragraph describing the threat model, [c], i.e., what specific real-world scenario this method is intended for.
>
> [a] We will make the abstract more quantitative, e.g.: "Empirically, we obtain a true positive rate of 96.7% at 0% false positive rate and watermark regurgitation rates exceeding 28% per document with only 40 watermarked documents."
>
> [b] We also agree that the threat model should be stated earlier and more explicitly, and will add it to the problem statement. Our setting is black-box dataset provenance auditing for fine-tuned LLMs, where a data owner proactively marks documents before release and later audits a suspicious fine-tuned model through API access only. The intended adversary is a model provider that may use the marked data for fine-tuning, but is not assumed to apply a removal pipeline specifically designed to defeat watermarks.
>
> [c] A practical scenario is trusted fine-tuning as a service, where a customer later tests whether marked documents were included in training. We will make this explicit in the introduction.
>
> > 2. [...] in the real world it is very common to apply necessary text filtering (e.g., removing invisible characters).[...]
>
> We agree that active removal of invisible characters is a limitation. However, Appendix E shows that under non-adversarial preprocessing the watermark is more robust than this may suggest. 5 out of 6 real-world preprocessing pipelines preserve all tested characters, and the remaining one removes only a small subset. Appendix E.4 further shows that tested tokenizers typically preserve the inserted characters as tokenizable signal. Such stripping affects completeness, not soundness: it may increase false negatives, but does not invalidate false-positive control.
>
> > 3. I do not understand the statement “we group them into syllables to improve memorization.” Why would LLMs memorize syllables more easily? Is there any experimental validation for this?
>
> Here, "syllable" denotes a short cluster of invisible characters, not a linguistic syllable. Empirically, individual insertion led to much lower regurgitation than shorter clusters, so we use clusters of length 4. We will clarify this in an empirical design choice.
>
> > 4. From Section 3.3, “A document D is split by spaces...”, it seems that this method would have difficulty handling languages such as Chinese and Japanese that do not use spaces. Does this paper implicitly restrict itself to English? This usage scenario should be clarified.
>
> We agree that splitting by spaces is a language-specific implementation detail. We will reformulate this step as splitting into meaningful text units prior to watermark insertion. For languages without spaces (e.g., Chinese or Japanese), standard segmentation tools such as jieba or fugashi can be applied before insertion.
>
> > 5. [a] The target models in the experiments are LLaMA-2-7b-hf and Mistral-7B-v0.1, both of which are somewhat outdated, and the fine-tuning datasets used in the experiments are also relatively small. [b] Why is the result for nbdocs = 50 on the News dataset missing?
>
>
> [a] To address this concern, we ran an additional experiment on a recent and larger model, GPT-OSS-20B, under a protocol aligned with Section 4.4 and using 100 watermarked documents per watermark:
>
> | dataset   | unique watermarks | regurgitation rates |
> |-----------|------------------|---------------------|
> | blog1k    | 5                | 0.40                |
> | blog1k    | 20               | 0.79                |
> | blog1k    | 30               | 0.67                |
> | blog1k    | 40               | 0.80                |
> | blog1k    | 50               | 0.78                |
> | poems     | 5                | 0.18                |
> | poems     | 20               | 0.43                |
> | poems     | 30               | 0.54                |
> | poems     | 40               | 0.68                |
> | poems     | 50               | 0.68                |
> | news  | 5                | 0.74                |
> | news  | 20               | 0.86                |
> | news  | 30               | 0.87                |
> | news  | 40               | 0.87                |
> | news  | 50               | 0.84                |
>
> While not a like-for-like replacement of the 7B experiments, these results show that the mechanism is not specific to smaller models. Larger models simply require more marked documents for stable memorization and detection.
>
> [b] LLaMA-2-7b-hf plateaued around nbdocs = 40, so later points were omitted for visual simplicity. We will add nbdocs = 50 to Figure 2 for completeness.

---

> > ### Author Rebuttal · Reviewer_T7RE · 2026-04-03
> >
> > I appreciate the authors effort in addressing the concerns. However, the filtering method is a simple countermeasure that could be plug in to the provided LLM. In this sense, I will consider maintaining my score.

---

> > > ### Author Response · Authors · 2026-04-06
> > >
> > > We thank the reviewer for their answer and we agree that filtering steps removing the inserted Unicode signal are a limitation, which we will make more explicit in the revision.
> > >
> > > A key distinction is between routine preprocessing introduced for standard collection or cleaning purposes, and filtering introduced specifically to suppress the audit signal. We agree that the latter can be technically simple once the scheme is known (it is also not entirely neutral in practice, since deliberately removing provenance markers may itself raise compliance or legal concerns in some settings). Our claim is therefore narrower: under proactive marking and the realistic non-targeted collection, preprocessing, and tokenization pipelines tested in Appendix E, the signal is preserved, enabling black-box provenance auditing with explicit false-positive control.
> > >
> > > The contribution is not to provide a tamper-proof mechanism, but to enable a reliable auditing signal under realistic black-box constraints. In this setting, practical value comes from providing sound evidence when the signal is present. Targeted removal may reduce completeness, potentially to zero, but it does not undermine the soundness of a positive finding when the signal is preserved.
> > >
> > > We will revise the paper to clarify this scope earlier and more explicitly.

---

### Decision · Program_Chairs · 2026-04-30

**Decision:**

Accept (regular)

**Comment:**

This paper introduces a novel, minimally invasive black-box auditing method to track data provenance in fine-tuned Large Language Models. By embedding a "cue-reply" structure using invisible Unicode characters, the method allows data owners to verify if their text was used in training.

The reviewers unanimously agreed that the paper addresses a highly important and timely problem. The proposed approach is technically sound, well-written, and introduces a clever mechanism that preserves the original text's visible semantics. Furthermore, the reviewers commended the statistically grounded decision procedure (the counterfactual ranking test), which provides a provable bound on the False Positive Rate.

**Resolution of Main Tensions:**
The primary point of debate during the review process centered on the method's practical robustness. Reviewers expressed valid concerns that the invisible Unicode watermarks are inherently fragile and could be easily defeated by simple text filtering, or suppressed by modern alignment techniques like RLHF.

However, the authors provided a highly effective rebuttal. They empirically demonstrated that their method survives the majority of standard, real-world data preparation pipelines (e.g., RedPajama, C4) in non-adversarial settings. The authors also transparently clarified the boundary conditions of their method, appropriately scoping it for Supervised Fine-Tuning (SFT) and honest-but-curious actors. While the method's fragility to an *aware* or *active* adversary remains a genuine limitation, this is a known dynamic within the watermarking literature. This limitation does not negate the paper's technical soundness, nor its utility as a foundational tool for proactive data auditing.

**Conclusion:**
This paper presents a rigorous, non-redundant contribution to the growing field of copyright and data provenance auditing. It will be highly useful to the ICML community. I recommend acceptance and encourage the authors to ensure their rebuttal clarifications regarding threat models and training regimes (SFT vs. RLHF) are heavily emphasized in the camera-ready version.